# Multimodal Aligned Semantic Knowledge for Unpaired Image-text Matching

**Laiguo Yin**[1], **Yixin Zhang**[2], **Yuqing Sun**[1]*, **Lizhen Cui**[1]*
[1]Shandong University [2]Nanyang Technological University
lgyin@mail.sdu.edu.cn, zhangyixin9610@gmail.com,
{sun_yuqing,clz}@sdu.edu.cn

## Abstract

While existing approaches address unpaired image-text matching by constructing cross-modal aligned knowledge, they often fail to identify semantically corresponding visual representations for Out-of-Distribution (OOD) words. Moreover, the distributional variance of visual representations associated with different words varies significantly, which negatively impacts matching accuracy. To address these issues, we propose a novel method namely Multimodal Aligned Semantic Knowledge (MASK), which leverages word embeddings as bridges to associate words with their corresponding prototypes, thereby enabling semantic knowledge alignment between the image and text modalities. For OOD words, the representative prototypes are constructed by leveraging the semantic relationships encoded in word embeddings. Beyond that, we introduce a prototype consistency contrastive learning loss to structurally regularize the feature space, effectively mitigating the adverse effects of variance. Experimental results on the Flickr30K and MSCOCO datasets demonstrate that MASK achieves superior performance in unpaired matching.

## 1 Introduction

Image–text matching has become an essential technique for various applications, such as visual question answering (Özdemir & Akagündüz, 2024; Lerner et al., 2024), image captioning (Fu et al., 2024; Wang et al., 2024a), cross-modal retrieval (Wang et al., 2024b; Li et al., 2024b) and so forth. Due to the heterogeneous representations and asymmetry of information between images and texts, accurately learning cross-modal semantic correspondences remains a challenging problem. Although training on large-scale paired image-text data has substantially improved matching accuracy, collecting and annotating such data at scale is often impractical in real-world scenarios.

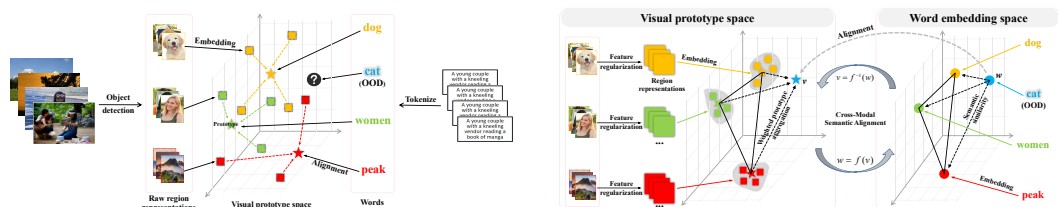

(a) Existing unpaired image-text matching      (b) Multimodal aligned semantic knowledge

Figure 1: Comparison between existing matching paradigms and our proposed unpaired framework.

To reduce the dependence of image-text matching on paired data, the unpaired image-text matching paradigm (Huang et al., 2022) was first proposed, in which domain-specific paired images and texts are assumed to be unavailable during model training. Inspired by the fact that the human brain can correlate arbitrary images and texts well while does not need to learn from large-scale paired

---

*Corresponding author.

images and texts, the unpaired image-text matching is implemented by modeling human brain-like knowledge, which is multimodal aligned and used to associate visual and linguistic information.

Note that unimodal visual or linguistic knowledge has been widely used for vision and language understanding task (Chen & Zhao, 2023). There are also some works (Li et al., 2025; Gao et al., 2025) directly combining these two types of knowledge together, but the resulting knowledge is not multimodal aligned. Another alternative is Multimodal Aligned Conceptual Knowledge (Huang et al., 2024a), which establishes correspondences between prototypical region representations and words, as shown in Figure 1 (a). However, these knowledge-based methods still face the following issues: **1) OOD words have not been thoroughly investigated.** Existing knowledge-based methods fail to leverage the underlying semantic structure to transfer the visual prototypes of known words to OOD words; **2) The influence of distributional variance has been largely overlooked.** The region representations corresponding to different words exhibit substantial appearance variations. Consequently, certain instances that deviate substantially from the distributional mean may be prone to misclassification into other words; **3) The raw region representation is insufficient in effectively capturing the semantic relationships between words.** The raw region representation is predominantly influenced by the co-occurrence relationships among regions. However, there is no inherent relationship between semantic relevance and co-occurrence patterns. For instance, while 'human' and 'hat' often co-occur in visual contexts, 'human' and 'gentleman' may exhibit a higher semantic similarity.

To address these issues, we propose a new method namely Multimodal Aligned Semantic Knowledge for unpaired image-text matching, which establishes semantic alignment between prototypical region representations and word embeddings, as shown in Figure 1 (b). We summarize our key contributions as follows:

- We propose a novel cross-modal semantic alignment method, MASK, which constructs representative prototypes for OOD words by exploiting the intrinsic relationships among word embeddings, thereby enhancing the model's generalization ability in unpaired image-text matching.

- We introduce a prototype consistency contrastive learning loss to structurally regularize the feature space, which explicitly encourages region representations associated with the same word to align closely with their prototype, thereby mitigating the adverse impact of distributional variance.

- We incorporate external knowledge from pre-trained word vectors as auxiliary supervision signals, which establishes a relation-preserving equivariant mapping between region representations and word embeddings, enabling the region representations to effectively capture semantic relationships among words.

## 2 RELATED WORK

### 2.1 MODEL-BASED MATCHING

Extensive model-based matching works have been made on measuring the semantic correlation between vision and language. To our knowledge, Socher *et al.* (Socher et al., 2013) might propose the first framework of Visual-Semantic Embedding (VSE) to correlate images and their class labels in a two-stream manner. Lee *et al.* (Lee et al., 2018) propose a Stacked Cross Attention Network(SCAN) to discover all latent alignments by using regions of the image and words in a sentence as context. The SCAN has been extensively studied from various aspects such as memory modeling (Huang et al., 2021), context modeling (Zhang et al., 2020) and graph structure (Liu et al., 2020). Later, by using millions or billions of paired images and texts for supervised model learning, many models (Li et al., 2020; Pan et al., 2023; Wu et al., 2024; Li et al., 2024a; Pham et al., 2024; Ge et al., 2024) based on multimodal versions of Transformer have been proposed and have achieved remarkable results. However, while these existing methods achieve relatively strong performance, they rely heavily on extensive paired image-text datasets for supervised training, which significantly restricts their applicability.

## 2.2 KNOWLEDGE-BASED MATCHING

Knowledge-based matching has been explored in a few vision and language understanding tasks. Feng *et al.* (Feng et al., 2019) use a visual concept detector to encourage generated captions to be semantically consistent with visual concepts. This work focuses more on the visual concepts while paying less attention to visual relations. Gu *et al.* (Gu et al., 2019) propose to align two scene graphs of images and texts, by mapping one to the other one, and vice versa. The performance of this work depends more on the accuracy of predicted scene graphs that contain both visual concepts and relations. Huang *et al.* (Huang et al., 2022) introduce Multimodal Aligned Conceptual Knowledge (MACK) for unpaired image-text matching, effectively addressing challenges associated with diverse appearances. Subsequently, Huang *et al.* (Huang et al., 2024b) further extend MACK to broaden its applicability. However, these approaches still face certain limitations, particularly in their inability to leverage the underlying semantic structure to transfer the prototype representations of known words to OOD words.

## 3 METHOD

This section illustrates the pipeline of obtaining the proposed MASK for unpaired image-text matching, as shown in Figure 2. For each region, the image embedding branch is utilized to extract region representations characterized by high cohesion and low coupling. For each word, the text embedding branch is employed to generate its corresponding word embedding. Consequently, we obtain a knowledge set in which each word is aligned with its corresponding prototypical region representation. This knowledge serves as a bridge in the knowledge-based image-text matching module, enabling the association between domain-specific images and texts and thereby supporting unpaired matching. It is worth noting that the knowledge can be fine-tuned to better adapt to specific domain (Appendix H). However, the fine-tuning step is optional according to whether unpaired data in certain domain are given or not.

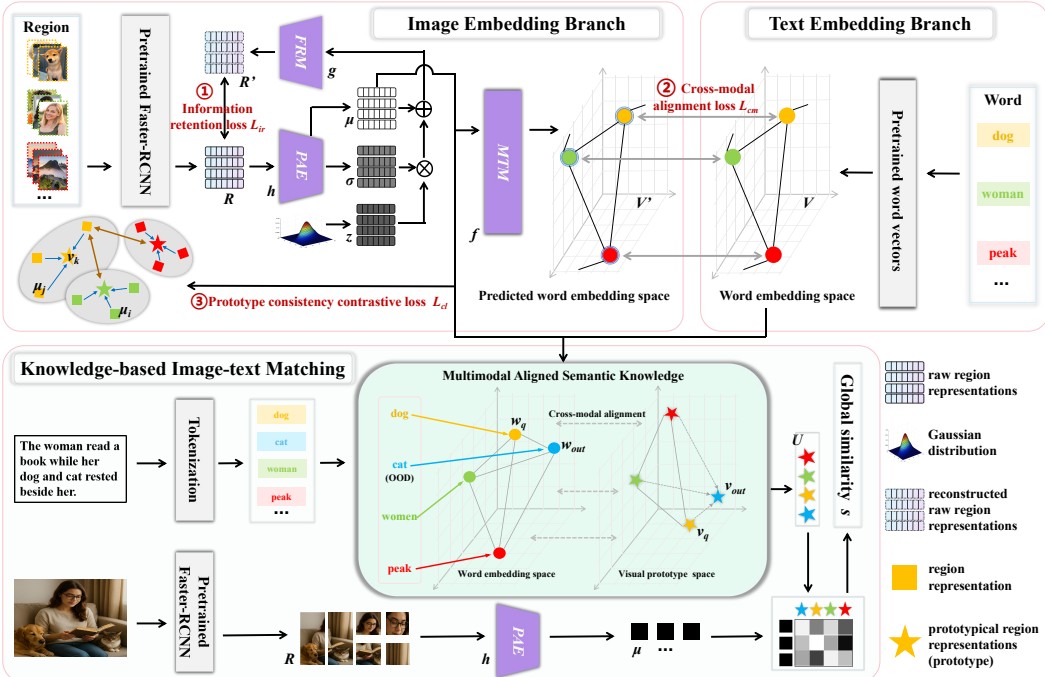

Figure 2: The Multimodal Aligned Semantic Knowledge (MASK) for unpaired image-text matching. The top figures illustrate how to obtain the knowledge and the bottom figures illustrate how to use the knowledge for unpaired image-text matching.

## 3.1 Multimodal Aligned Semantic Knowledge

In addition to the semantic concepts, the studied knowledge also has another important property of cross-modal one-to-one alignment. For each word, its semantically related objects in different regions often exhibit diverse visual appearances, which could easily lead to confusion in practice. Therefore, rather than align each word to multiple related regions in an one-to-many manner, the MASK aligns each word to a single prototypical region, with the goal to alleviate the issue of appearance variation. In particular, we formulate the knowledge as a set of semantic concepts having paired multimodal representations $\{(w_k, v_k)\}_{k=1,\ldots,K}$, where $w_k$ and $v_k$ are the word embedding and prototypical region representation of the $k$-th semantic concept, respectively, and $K$ is the total number of semantic concepts.

As shown in Figure 2, for each word, we compute the word embeddings $w_k$ by using pre-trained word vectors. For each region, we first extract the raw region representations $r_j (j = 1, \ldots, J_k)$ by feeding a bounding box and an image into the pre-trained object detection model Faster-RCNN. Then we extract region representations $\mu_j$ by utilizing the Prototype-Aware Encoder ($PAE$) $h$, with $r_j$ as the input:

$$\mu_j, \sigma_j = h(r_j; \Theta_h), \tag{1}$$

where $\Theta_h$ is the parameters of $h$ and $\sigma_j$ represents the variance of distribution. Finally, we compute the prototypical region representations $v_k$ by averaging all related region representations $\{\mu_j\}_{j=1,\ldots,J_k}$:

$$v_k = \frac{1}{J_k} \sum_{j=1}^{J_k} \mu_j, \tag{2}$$

where the $J_k$ indicates the number of regions for the $k$-th semantic concept.

## 3.2 Image Embedding Branch

Given a batch $B$ of paired regions and words, we first obtain raw region representations $R = \{r_n\}_{n=1,\ldots,B}$ ($R \in \mathcal{R}^{B \times M}$), where $M$ is the dimension of $r_n$. We then extract region representations $\mu$ using the $PAE$ model $h$, which takes $R$ as input and consists of a Fully Connected (FC) layer followed by three self-attention layers:

$$\mu, \sigma = h(R; \Theta_h), \tag{3}$$

where the mean $\mu \in \mathcal{R}^{B \times Z}$ and variance $\sigma \in \mathcal{R}^{B \times Z}$ are used to preserve the information of the raw region representations $R$ by using the Feature Restoration Module ($FRM$) $g$ comprising a self-attention layer and two FC layers:

$$R' = g(\mu, \sigma, z; \Theta_g), \tag{4}$$

where the $z$ is a random vector sampled from a standard normal distribution and $\Theta_g$ is the parameters of $g$. The $Z$ represents the feature dimension of latent space. As shown in Figure 2, the $PAE$ model $h$ and $FRM$ model $g$ are trained jointly using the **i**nformation **r**etention loss function $\mathcal{L}_{ir}$:

$$\mathcal{L}_{ir} = D_{KL}( N(\mu, \sigma^2) \,||\, N(0, 1) ) + \mathbb{E}_{(r_n, r'_n) \sim (R, R')} [\|r_n - r'_n\|_2^2], \tag{5}$$

where the $D_{KL}( N(\mu, \sigma^2) \,||\, N(0, 1) )$ implies that the data distribution in the latent space gradually approaches the standard normal distribution. The $\mathbb{E}_{(r_n, r'_n) \sim (R, R')} [\|r_n - r'_n\|_2^2]$ measures the difference between the reconstructed raw region representations $R'$ and the raw region representations $R$. The loss $\mathcal{L}_{ir}$ ensures that the mean $\mu$ retains a significant amount of information from the raw region representations $R$.

Inspired by clustering theory and contrastive learning, we design a prototype consistency **c**ontrastive **l**earning loss $\mathcal{L}_{cl}$ to reduce the influence of distributional variance between prototypes and their related region representations. The loss $\mathcal{L}_{cl}$ employs prototypes as class centers, maximizing the similarity between region representations and their corresponding prototypes while minimizing similarity with other prototypes, thereby achieving intra-class aggregation and inter-class separation. Compared to traditional instance-to-instance contrastive learning, $\mathcal{L}_{cl}$ introduces prototypes as global semantic representatives, explicitly aggregating instances of the same class around their corresponding prototypes. This process constructs a more structured and discriminative representation

space, enabling the model to capture clearer semantic boundaries. The loss $\mathcal{L}_{cl}$ is defined as follows (Appendix G):

$$\mathcal{L}_{cl} = -\frac{1}{B}\sum_{k=1}^{B} log \frac{exp(v_k \cdot \mu_+/\tau)}{\sum_{n=1}^{B} exp(v_k \cdot \mu_n/\tau)}, \tag{6}$$

where the $\mu_+$ refers to the region representations associated with the prototypical region representations $v_k$, i.e., positive examples. The hyperparameter $\tau$ regulates the capacity of the model to distinguish between negative examples. The loss $\mathcal{L}_{cl}$ encourages all region representations corresponding to the same word to be closer to each other, while driving the region representations corresponding to different words farther apart, which effectively mitigates the impact of variance among different words on the similarity computation.

### 3.3 TEXT EMBEDDING BRANCH

Given a batch $B$ of paired regions and words, we obtain word embeddings $V = \{w_k\}_{k=1,...,B}$ ($V \in \mathcal{R}^{B \times N}$) by utilizing the pretrained word vectors, where $N$ is the dimension of $w_k$. Pre-trained word embeddings typically exhibit well-structured semantic properties, where semantically related words are mapped to vectors that are close to each other in the embedding space. To enable region representations to effectively capture semantic correlations between words, we utilize a Modality Transfer Model ($MTM$) $f$ with three self-attention layers and three FC layers that can map the mean $\mu$ output by the $PAE$ model $h$ into the word embedding space:

$$V' = f(\mu;\ \Theta_f), \tag{7}$$

where the $\Theta_f$ is the parameters of the $MTM$ model $f$ and $V' \in \mathcal{R}^{B \times N}$ represents the predicted word embeddings. The model $f$ is a relation-preserving equivariant mapping that lays the foundation for constructing prototypical region representations corresponding to OOD words. Formally, for any two region representations $\mu_i$ and $\mu_j$, the function $f$ should satisfy (Appendix B):

$$d_s(f(\mu_i;\ \Theta_f),\ f(\mu_j;\ \Theta_f)) \propto d_s(\mu_i,\ \mu_j), \tag{8}$$

where the distance metric $d_s$ captures the pairwise relations between representations within each modality. The $PAE$ model $h$ and $MTM$ model $f$ are trained jointly using the **c**ross-**m**odal alignment loss function $\mathcal{L}_{cm}$ (Appendix C, E and F):

$$\mathcal{L}_{cm} = \mathbb{E}[(1 - cos(\frac{w_i}{\|w_i\|_2}, \frac{w_i'}{\|w_i'\|_2}))] + \mathbb{E}[((cos(\frac{w_i'}{\|w_i'\|_2}, \frac{w_j'}{\|w_j'\|_2}) - cos(\frac{\mu_i}{\|\mu_i\|_2}, \frac{\mu_j}{\|\mu_j\|_2})))^2], \tag{9}$$

where $w_i', w_j' \in V'(i \neq j)$ and $w_i \in V$. The loss function $\mathcal{L}_{cm}$ enforces the predicted word embeddings $V'$ to gradually converge toward the word embeddings $V$, while simultaneously ensuring that the region representations effectively capture the semantic relationships between words.

### 3.4 KNOWLEDGE-BASED IMAGE-TEXT MATCHING

To decide whether a given image and a text are matched or not, we first obtain a set of raw region representations $R = \{r_i\}_{i=1,...,I}$ ($R \in \mathcal{R}^{I \times M}$) using the Faster-RCNN above and a set of parsed words through tokenization operation implemented via NLTK [1], as shown in Figure 2. Then, we use the knowledge as a cross-modal bridge to represent all the words into the corresponding prototypical region representations $U = \{v_j\}_{j=1,...,J}$ ($U \in \mathcal{R}^{J \times Z}$). For the set of regions $R$, we extract region representations $\mu \in \mathcal{R}^{I \times Z}$ by utilizing the $PAE$ model $h$. Finally, we obtain the desired global similarity score $s$ for the given image and text as:

$$s = \rho(\ \mu \cdot U^T\ ), \tag{10}$$

where $\rho(\cdot)$ denotes the max-mean pooling operation, which first performs max pooling along the column dimension and then mean pooling along the row dimension of the input matrix.

However, the scope of knowledge is inherently limited and heavily reliant on the volume of paired data available in public datasets. The vocabulary size supported by the pre-trained word vectors

---

[1]https://www.nltk.org/

significantly surpasses the scale of the existing knowledge. Therefore, for OOD words relative to the knowledge, their corresponding word embeddings can typically be obtained by leveraging pre-trained word vectors. To fully utilize these OOD words, we first sample $m$ paired multimodal representations $\{(w_q, v_q)\}_{q=1}^m$ from the knowledge. Then, we calculate the similarity scores $\{s_q\}_{q=1}^m$ between the $m$ word embeddings $\{w_q\}_{q=1}^m$ and the word embedding $w_{out}$:

$$\{s_q\}_{q=1}^m = softmax(w_{out} \cdot \{w_q\}_{q=1}^m). \tag{11}$$

By utilizing the sampled prototypical region representations $\{v_q\}_{q=1}^m$ as base vectors and the similarity scores $\{s_q\}_{q=1}^m$, we can obtain the prototypical region representation $v_{out}$ corresponding to word embedding $w_{out}$:

$$v_{out} = \sum_{q=1}^m s_q \cdot v_q. \tag{12}$$

In unpaired image-text matching, constructing prototypical region representations based on semantic similarities between words enables the effective utilization of information from OOD words.

To ensure the semantic quality of the visual prototypes constructed for OOD words, we select the top-$m$ paired multimodal representations from the knowledge whose word embeddings are most relevant to OOD words. This selection strategy is motivated by the local linearity property of word embeddings on the semantic manifold. Semantically related words lie close to each other in the embedding space and approximately reside in a locally linear subspace. Consequently, the top-$m$ neighbors provide the most informative directions for reconstructing the corresponding visual representations. Moreover, the $\mathcal{L}_{cm}$ constrains local alignment between the word embedding space and the visual prototype space, making nearest neighbors in the embedding space more likely to preserve geometric relationships in the prototype space, thereby reducing reconstruction bias. In this way, the top-$m$ neighbors effectively capture the most salient semantic and structural information needed for accurate and robust prototype estimation. To obtain these top-$m$ semantic neighbors for OOD words, we first normalize all word embeddings $\{w_q\}_{q=1}^K$ in the knowledge, and denote the normalized embeddings as $\{\frac{w_q}{\|w_q\|}\}_{q=1}^K$. Similarly, the normalized embedding of OOD words as $\frac{w_{out}}{\|w_{out}\|}$. The similarity between $\frac{w_{out}}{\|w_{out}\|}$ and $\{\frac{w_q}{\|w_q\|}\}_{q=1}^K$ are computed as:

$$\{s_q\}_{q=1}^K = \frac{w_{out}}{\|w_{out}\|} \cdot \{\frac{w_q}{\|w_q\|}\}_{q=1}^K. \tag{13}$$

The top-$m$ nearest neighbors $\{s_q\}_{q=1}^m$ are then selected based on $\{s_q\}_{q=1}^K$ to support subsequent visual prototype construction.

### 3.5 MODEL TRAINING

The studied knowledge mainly contains dataset-independent semantic concepts, with the goal to be generally applicable to different scenarios. The semantic concepts are multimodal, which includes objects and attributes in images, and nouns and adjectives in texts. To obtain them, we resort to publicly available dataset Visual Genome (VG) (Krishna et al., 2017) [2] and collect corresponding words and regions. For the textual knowledge, we obtain various words from synsets in the dataset. For the visual knowledge, we detect regions from images and then associate them with the words.

After collecting a set of words and their semantically related image regions from publicly available datasets, we train our model on these paired data to construct MASK. The loss of the entire training process is expressed as $\mathcal{L}$:

$$\mathcal{L} = \mathcal{L}_{ir} + \lambda_1 \mathcal{L}_{cm} + \lambda_2 \mathcal{L}_{cl}, \tag{14}$$

where $\lambda_1$ and $\lambda_2$ are trade-off factors for balancing different losses. By optimizing the loss $\mathcal{L}$, the region representations exhibit properties of high cohesion and low coupling, indicating that representations corresponding to the same word become more compact and semantically consistent.

### 3.6 RE-RANKING EXTENSION

The proposed MASK is a knowledge-based approach, which differs significantly from existing data-driven models. Due to this distinction, it is expected to exhibit complementary properties when

---

[2] http://visualgenome.org/

combined with existing models. We extend MASK into a re-ranking method to re-rank the initial results produced by existing multimodal models.

Taking the sub-task of image retrieval as an example, given a text query and a gallery of $L$ images, an existing model can compute similarity scores and produce a similarity vector $\tilde{s} \in \mathcal{R}^{L \times 1}$. By sorting the values of $\tilde{s}$ in descending order, the model ranks the images and identifies the top-$k$ candidates, denoted as $\tilde{s}^k \in \mathcal{R}^{k \times 1}$. For the text query and the top-$k$ retrieved images, we then compute an additional similarity vector $s^k \in \mathcal{R}^{k \times 1}$ using MASK in an unpaired image-text matching setting. Finally, the two similarity vectors are combined using a balancing factor $\alpha$:

$$\hat{s}^k = ZS(\tilde{s}^k) + \alpha \cdot ZS(s^k), \tag{15}$$

where $ZS$ represents the Z-Score normalization and $\hat{s}^k$ is the new similarity vector that can be used to re-rank the top-$k$ images to improve the rank of matched images.

## 4 EXPERIMENTS

### 4.1 DATASETS AND METRICS

We test the performance of MASK on two standard datasets: Flickr30K and MSCOCO. The commonly used evaluation criterions are "R@1", "R@5" and "R@10", i.e., recall rates at the top-1, 5 and 10 results. Following existing works (Ge et al., 2024), we use an additional criterion of "Rs" by summing all the recall rates to evaluate the overall performance. Experimental details are provided in the Appendix I.2.

### 4.2 UNPAIRED IMAGE-TEXT MATCHING

We design several experiments comparing state-of-the-art model-based matching methods (e.g., $CHAN$ (Pan et al., 2023), $DSRLN$ (Wu et al., 2024), $CORA$ (Pham et al., 2024), $BOOM$ (Li et al., 2024a), and $3SHNet$ (Ge et al., 2024)) and knowledge-based matching methods (e.g., $MACK$ (Huang et al., 2022) and $MACK^{VG-M}$ (Huang et al., 2024b)) to verify the effectiveness for unpaired image-text matching. We compare the performance of unpaired image-text matching in Table 1. Model-based matching and knowledge-based matching exhibit comparable performance on the Flickr30K dataset, whereas their performance diverges significantly on the MSCOCO dataset. Compared to Flickr30K, MSCOCO exhibits greater sample diversity, with images typically containing multiple target objects and semantic regions, resulting in more complex visual structures. The knowledge-based matching constructs explicit multimodal-aligned knowledge as a bridge between regions and words, facilitating more accurate modeling of local visual-semantic relationships in complex visual scenes.

Table 1: Performance comparison between model-based matching and knowledge-based matching on the Flickr30K and MSCOCO datasets for the unpaired image-text matching.

| | Method | Flickr30K dataset | | | | | | | MSCOCO dataset | | | | | | |
| --- | --- | --- | --- | --- | --- | --- | --- | --- | --- | --- | --- | --- | --- | --- | --- |
| | | Image Retrieval | | | Image Annotation | | | Rs | Image Retrieval | | | Image Annotation | | | Rs |
| | | R@1 | R@5 | R@10 | R@1 | R@5 | R@10 | | R@1 | R@5 | R@10 | R@1 | R@5 | R@10 | |
| model-based matching | $CHAN_{2023}$ | 2.2 | 10.4 | 16.9 | 1.5 | 7.8 | 24.5 | 63.3 | 2.1 | 9.1 | 15.7 | 3.5 | 11.9 | 22.5 | 64.8 |
| | $DSRLN_{2024}$ | 4.3 | 14.1 | 22.1 | 7.0 | 21.9 | 37.8 | 107.2 | 4.4 | 14.6 | 21.7 | 5.5 | 20.0 | 39.0 | 105.2 |
| | $CORA_{2024}$ | 3.6 | 12.0 | 22.9 | 8.3 | 22.7 | 33.9 | 103.4 | 5.5 | 18.7 | 32.5 | 12.4 | 29.3 | 45.2 | 143.6 |
| | $BOOM_{2024}$ | 3.9 | 12.6 | **23.8** | 8.3 | 22.6 | 35.2 | 106.4 | 5.8 | 19.2 | 32.6 | 12.9 | 29.8 | 45.4 | 145.7 |
| | $3SHNet_{2024}$ | 3.8 | 12.2 | 23.1 | 8.1 | 22.0 | 34.3 | 103.5 | 6.0 | 19.7 | 33.4 | 13.2 | 29.6 | 47.8 | 149.7 |
| knowledge-based matching | $MACK_{2022}$ | 3.0 | 9.9 | 15.4 | 10.1 | 24.6 | 32.3 | 95.3 | 7.2 | 25.9 | 40.6 | 21.8 | 46.2 | 60.0 | 201.7 |
| | $MACK^{VG-M}_{2024}$ | 3.8 | 11.3 | 17.4 | 10.4 | 26.8 | 35.1 | 104.8 | 7.2 | 25.2 | 41.4 | 21.9 | 46.6 | **62.9** | 205.2 |
| | **MASK** | **4.8** | **14.8** | 22.0 | **12.1** | **30.1** | **39.0** | **122.8** | **7.6** | **26.7** | **41.8** | **22.7** | **48.5** | 62.2 | **209.5** |
| -w/o region prototypes | $MACK_{2022}$ | 1.6 | 5.3 | 8.8 | 5.3 | 16.4 | 22.7 | 60.1 | 4.6 | 17.0 | 28.2 | 13.3 | 31.7 | 44.2 | 139.0 |
| | $MACK^{VG-M}_{2024}$ | 1.7 | 7.0 | 10.9 | 5.3 | 16.2 | 23.4 | 64.5 | 4.6 | 16.9 | 29.6 | 13.3 | 31.4 | 45.3 | 141.1 |
| | **MASK** | **2.5** | **9.1** | **14.5** | **5.6** | **17.1** | **23.9** | **72.7** | **5.2** | **18.9** | **31.7** | **13.6** | **33.5** | **45.6** | **148.5** |
| -w/o max-mean pooling | $MACK_{2022}$ | 2.5 | 9.0 | 14.3 | 1.7 | 5.6 | 8.6 | 41.7 | 6.3 | 21.4 | 33.3 | 3.9 | 11.6 | 17.6 | 94.1 |
| | $MACK^{VG-M}_{2024}$ | 2.7 | 9.4 | 17.2 | 1.7 | 5.9 | 10.3 | 47.2 | 6.7 | 22.9 | 36.1 | 3.8 | 11.6 | 19.2 | 100.3 |
| | **MASK** | **4.5** | **14.1** | **22.1** | **2.9** | **7.7** | **12.0** | **63.3** | **8.0** | **26.0** | **40.2** | **4.8** | **12.8** | **20.9** | **112.7** |

The region prototypes and max-mean pooling have a significant impact on knowledge-based unpaired matching methods. However, MASK consistently outperforms existing methods. This is primarily because the MASK exhibits a strong intra-class cohesion among region representations, i.e., the variance between any region representation and the prototypical region representation is relatively small. Consequently, replacing the prototypical region representation with a randomly

selected region representation has a minimal impact on overall performance. Furthermore, incorporating semantic relationships between word embeddings reduces the coupling among region representations across words. Therefore, substituting max-mean pooling with global mean has a minor effect on overall performance.

## 4.3 Zero-shot Image-text Matching

To evaluate the effectiveness of the MASK in complementing with pre-trained models for zero-shot image-text matching, we compare several state-of-the-art re-ranking strategies, including $MACK$ (Huang et al., 2022), $LeaPRR$ (Qu et al., 2023), $MACK^{VG-M}$ (Huang et al., 2024b) and $FR$ (Wei et al., 2025). The original and re-ranked performance are compared in Table 2. We can see that although the original accuracies are already high, further improvements can be achieved by applying re-ranking strategies to these pre-trained models. Among them, MASK yields more substantial performance gains. This can be attributed to the high cohesion and low coupling of the region representations, which ensure that each region representation remains closest to its corresponding prototypical region representation while maintaining substantial spatial separation from those of non-corresponding terms. Consequently, MASK reduces the risk of region representations being misclassified in zero-shot image-text matching. These evidences demonstrate that the MASK can be well combined with existing models to further improve their performance.

Table 2: Zero-shot image-text matching by re-ranking two state-of-the-art models on the Flickr30K and MSCOCO datasets.

| Method | Flickr30K dataset | | | | | | | MSCOCO dataset | | | | | | |
| | Image Retrieval | | | Image Annotation | | | Rs | Image Retrieval | | | Image Annotation | | | Rs |
| | R@1 | R@5 | R@10 | R@1 | R@5 | R@10 | | R@1 | R@5 | R@10 | R@1 | R@5 | R@10 | |
| $CLIP$ | 65.4 | 87.2 | 91.7 | 85.4 | 97.1 | 98.7 | 525.5 | 35.3 | 60.0 | 70.1 | 55.2 | 78.7 | 86.7 | 386.0 |
| $CLIP + MACK_{2022}$ | 66.8 | 88.2 | 92.6 | 86.2 | 97.2 | 98.9 | 529.9 | 36.9 | 61.6 | 71.7 | 55.7 | 79.6 | 87.1 | 392.6 |
| $CLIP + LeaPRR_{2023}$ | 65.9 | 87.9 | 92.8 | 86.2 | 97.3 | 98.1 | 528.2 | 36.2 | 61.9 | 70.8 | 55.9 | 80.2 | 85.4 | 390.4 |
| $CLIP + MACK^{VG-M}{}_{2024}$ | 66.9 | 88.4 | 92.8 | 87.6 | 96.9 | 99.0 | 531.6 | 37.2 | 62.0 | 71.8 | 56.3 | 80.0 | 87.1 | 394.4 |
| $CLIP + FR_{2025}$ | 66.4 | 88.4 | 93.1 | 86.8 | 97.3 | **99.1** | 531.1 | 36.7 | 62.1 | 72.4 | 56.2 | 80.3 | 87.8 | 395.5 |
| $CLIP + $ **MASK** | **67.3** | **88.7** | **93.9** | **87.9** | **97.6** | 98.9 | **534.3** | **37.6** | **62.7** | **73.9** | **56.7** | **80.8** | **88.7** | **400.4** |
| $ALBEF$ | 59.9 | 84.8 | 90.5 | 78.2 | 95.5 | **97.9** | 506.9 | 40.2 | 68.4 | 78.9 | 62.4 | 85.9 | 92.1 | 428.3 |
| $ALBEF + MACK_{2022}$ | 61.8 | 85.8 | 91.3 | 80.1 | 96.4 | 97.7 | 513.1 | 41.0 | 69.0 | 79.4 | 62.4 | 86.1 | **92.7** | 430.9 |
| $ALBEF + LeaPRR_{2023}$ | 61.4 | 85.9 | 91.6 | 79.3 | 96.2 | 97.5 | 511.9 | 40.7 | 69.1 | 79.6 | 62.8 | 86.4 | 91.8 | 430.4 |
| $ALBEF + MACK^{VG-M}{}_{2024}$ | 61.8 | 86.1 | 91.6 | 80.5 | 95.5 | **97.9** | 513.4 | 41.6 | 69.5 | 79.7 | 63.0 | 86.0 | 92.4 | 432.2 |
| $ALBEF + FR_{2025}$ | 61.7 | 86.4 | 91.8 | 80.8 | 96.2 | 97.6 | 514.5 | 41.9 | 69.8 | 80.1 | 62.8 | 86.3 | 92.1 | 433.0 |
| $ALBEF + $ **MASK** | **63.1** | **86.6** | **92.0** | **81.6** | **96.9** | 97.9 | **518.1** | **42.1** | **70.5** | **80.8** | **63.6** | **86.9** | 92.3 | **436.2** |

## 4.4 Knowledge Visualization

To qualitatively illustrate major differences between MACK and MASK, we visualize two low-dimensional word distributions in Figure 3. The words in the four numbered groups (marked by dashed lines in different colors) are about animals, transports, faces and humans, respectively. In the left distribution corresponding to MACK, there are still some related words that have remote distances. In contrast, the right distribution generated by MASK is semantically more compact. The underlying mechanism is that semantic relationships between word embeddings are incorporated during model training, ensuring that the corresponding prototypical region representations also exhibit semantic associations. These evidences indicate that the MASK can make their prototypical region representations more discriminative.

## 4.5 OOD Words Analysis and Loss Ablation Analysis

To evaluate the impact of OOD words on image-text matching accuracy, we conduct a comparative experiment in Table 3. We observe that image-text matching accuracy significantly improves in both image retrieval and image annotation tasks when OOD words are incorporated, and this improvement is consistently validated across different datasets. Therefore, leveraging the semantic relationships between OOD words and known words to construct corresponding prototypical region representations for OOD words is an effective approach. This phenomenon can be attributed to the relation-preserving equivariant mapping, as shown in Eq (8). As a result, region representations inherit the semantic structure encoded in the word embeddings, allowing the relationships between regions to reflect semantic distances and similarities, thereby enhancing the generalization ability of the matching process.

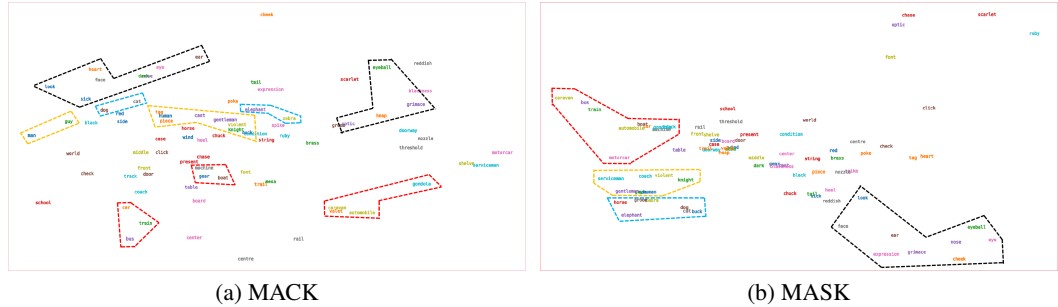

| (a) MACK | (b) MASK |

Figure 3: Visualization of prototypical region representations from MACK and MASK. Each word indicates its corresponding prototypical region representation embedded by t-SNE. In the two word distributions, we group semantically related words by using same-colored dashed lines.

Table 3: Ablation of the overall loss and the impact of OOD words on unpaired image-text matching.

| Method | Flickr30K dataset | | | | | | | MSCOCO dataset | | | | | | |
|---|---|---|---|---|---|---|---|---|---|---|---|---|---|---|
| | Image Retrieval | | | Image Annotation | | | Rs | Image Retrieval | | | Image Annotation | | | Rs |
| | R@1 | R@5 | R@10 | R@1 | R@5 | R@10 | | R@1 | R@5 | R@10 | R@1 | R@5 | R@10 | |
| $MASK$ | **4.8** | **14.8** | **22.0** | **12.1** | **30.1** | **39.0** | **122.8** | **7.6** | **26.7** | **41.8** | **22.7** | **48.5** | **62.2** | **209.5** |
| $MASK_{w/o\ OODwords}$ | 4.5 | 13.6 | 20.5 | 11.8 | 28.5 | 37.3 | 116.2 | 7.2 | 24.5 | 38.2 | 22.2 | 44.6 | 56.4 | 193.1 |
| $MASK$ | **4.8** | **14.8** | **22.0** | **12.1** | **30.1** | **39.0** | **122.8** | **7.6** | **26.7** | **41.8** | **22.7** | **48.5** | **62.2** | **209.5** |
| $MASK_{w/o\ \mathcal{L}_{cm}}$ | 3.1 | 10.5 | 15.2 | 10.2 | 26.6 | 35.4 | 101.0 | 5.2 | 16.5 | 24.9 | 17.2 | 37.6 | 49.4 | 150.8 |
| $MASK_{w/o\ \mathcal{L}_{cl}}$ | 2.6 | 8.3 | 13.7 | 9.1 | 23.9 | 34.8 | 92.4 | 4.5 | 13.6 | 20.9 | 13.3 | 31.0 | 40.2 | 123.5 |

To testify the contribution of each component of loss $\mathcal{L}$ to overall performance, we compare the performance of various losses in Table 3. It can be observed that the performance of the overall loss function $\mathcal{L}$ deteriorates when certain components are omitted. Among them, the prototype consistency contrastive learning loss $\mathcal{L}_{cl}$ makes the largest contribution to the performance. It is reasonable since the loss $\mathcal{L}_{cl}$ constrains all region representations related the same word to be close to each other during model training. Additionally, the semantic relationships between word embeddings in loss $\mathcal{L}_{cm}$ serve only as a reference for determining the degree of separation between region representations. Therefore, the loss $\mathcal{L}_{cl}$ can make the prototypical region representations more discriminative, which obviously affects the accuracy of matching.

## 4.6 HYPERPARAMETER ANALYSIS

The proposed MASK involves two trade-off parameters $\lambda_1$ and $\lambda_2$ in Eq.(14). To evaluate the impact of different hyperparameters on matching accuracy, we design three controlled experiments in Table 4. The results indicate that MASK achieves the best performance in unpaired image-text matching at $\lambda_1 = \lambda_2$. Compared to $\frac{\lambda_1}{\lambda_2} = 3.0$ (i.e., $\lambda_1 > \lambda_2$), the $\frac{\lambda_1}{\lambda_2} = 0.3$ achieves better performance, with improvements of approximately 3.5% and 10.3% in Rs on Flickr30K and MSCOCO, respectively. These findings suggest that the cross-modal alignment loss and prototype consistency contrastive loss are complementary, each contributing distinct yet essential benefits to the overall performance. By jointly optimizing these two losses in a balanced manner, we mitigate the risk of overfitting to a single loss term, thereby improving the model's generalization capability.

Table 4: Unpaired image-text matching by MASK using different $\lambda_1/\lambda_2$ on the Flickr30K and MSCOCO datasets.

| $\frac{\lambda_1}{\lambda_2}$ | Flickr30K dataset | | | | | | | MSCOCO dataset | | | | | | |
|---|---|---|---|---|---|---|---|---|---|---|---|---|---|---|
| | Image Retrieval | | | Image Annotation | | | Rs | Image Retrieval | | | Image Annotation | | | Rs |
| | R@1 | R@5 | R@10 | R@1 | R@5 | R@10 | | R@1 | R@5 | R@10 | R@1 | R@5 | R@10 | |
| 0.3 | 4.2 | 13.0 | 19.2 | 11.6 | 27.9 | 36.8 | 112.7 | 6.8 | 22.9 | 35.5 | 20.9 | 43.2 | 55.8 | 185.1 |
| 1.0 | **4.8** | **14.8** | **22.0** | **12.1** | **30.1** | **39.0** | **122.8** | **7.6** | **26.7** | **41.8** | **22.7** | **48.5** | **62.2** | **209.5** |
| 3.0 | 3.9 | 12.2 | 18.0 | 11.2 | 27.5 | 36.4 | 109.2 | 6.3 | 21.0 | 32.3 | 19.8 | 41.5 | 53.9 | 174.8 |

## 4.7 ANALYSIS OF SAMPLING SIZE

The sampling size $m$ for selecting paired multimodal representations critically affects the semantic quality of the constructed prototypes. To evaluate the impact of different sampling sizes on matching accuracy, we perform the experiment of unpaired image-text matching by MASK using different sampling sizes $m$ on the Flickr30K and MSCOCO datasets in Table 5. Experimental results show that the matching accuracy follows a rise–then–fall trend as the sampling size $m$ increases, achieving its optimum around $m = 50$. This phenomenon can be explained as follows. When the sampling size is too small, the constructed visual prototype relies excessively on only a few nearest neighbors. Although this preserves strong local semantic characteristics, it also makes the prototype highly sensitive to noise and outliers in the word embedding space. As the sampling size increases to a moderate level, more semantically relevant neighbors contribute their visual information, thereby enhancing robustness and discriminability. However, when the sampling size becomes too large, semantically weak or marginal neighbors begin to dominate. Their less relevant visual cues dilute the contributions of the core semantic neighbors, ultimately reducing matching accuracy.

Table 5: Unpaired image-text matching by MASK using different sampling sizes $m$ on the Flickr30K and MSCOCO datasets.

| $m$ | Flickr30K dataset | | | | | | | MSCOCO dataset | | | | | | |
| | Image Retrieval | | | Image Annotation | | | Rs | Image Retrieval | | | Image Annotation | | | Rs |
| | R@1 | R@5 | R@10 | R@1 | R@5 | R@10 | | R@1 | R@5 | R@10 | R@1 | R@5 | R@10 | |
|---|---|---|---|---|---|---|---|---|---|---|---|---|---|---|
| 1 | 4.6 | 13.9 | 20.9 | 11.9 | 28.9 | 37.7 | 117.9 | 7.3 | 25.1 | 39.1 | 22.3 | 45.6 | 57.9 | 197.3 |
| 10 | 4.8 | 14.8 | 22.0 | 12.1 | 30.1 | 39.0 | 122.8 | 7.6 | 26.7 | 41.8 | 22.7 | 48.5 | 62.2 | 209.5 |
| 50 | **5.0** | **15.8** | **23.2** | **12.3** | **31.4** | **40.4** | **128.1** | **7.9** | **28.5** | **44.7** | **23.1** | **51.6** | **66.8** | **222.6** |
| 100 | 4.9 | 15.3 | 22.6 | 12.2 | 30.7 | 39.7 | 125.4 | 7.7 | 27.6 | 43.3 | 22.9 | 50.0 | 64.5 | 216.0 |

## 4.8 RE-RANKING ANALYSIS

An important balancing factor $\alpha$, is introduced for re-ranking the existing models in Eq.(15). To investigate the impact of $\alpha$ on final performance, we conduct zero-shot image-text matching experiments using MASK (CLIP) in Table 6. The results show that the best performance on both datasets is obtained at $\alpha = 0.15$. When $\alpha$ is relatively small, the re-ranking method yields limited improvements, as matching performance is largely dominated by the pre-trained CLIP model. In contrast, as $\alpha$ increases, the influence of the MASK on matching performance becomes more pronounced. Since the knowledge is not learned from domain-specific paired image-text data, potential distributional discrepancies may arise, which can adversely affect the overall performance. Therefore, an appropriate value of $\alpha$ must be carefully chosen to achieve an optimal balance between semantic alignment capability and generalization performance.

Table 6: Zero-shot image-text matching by MASK (CLIP) using different $\alpha$ on the Flickr30K and MSCOCO datasets.

| $\alpha$ | Flickr30K dataset | | | | | | | MSCOCO dataset | | | | | | |
| | Image Retrieval | | | Image Annotation | | | Rs | Image Retrieval | | | Image Annotation | | | Rs |
| | R@1 | R@5 | R@10 | R@1 | R@5 | R@10 | | R@1 | R@5 | R@10 | R@1 | R@5 | R@10 | |
|---|---|---|---|---|---|---|---|---|---|---|---|---|---|---|
| 0 | 65.4 | 87.2 | 91.7 | 85.4 | 97.1 | 98.7 | 525.5 | 35.3 | 60.0 | 70.1 | 55.2 | 78.7 | 86.7 | 386.0 |
| 0.1 | 67.1 | **88.7** | 93.7 | 87.2 | 97.3 | **99.1** | 533.1 | 37.1 | 62.3 | 73.6 | 56.3 | 80.3 | 88.6 | 398.2 |
| 0.15 | **67.3** | **88.7** | **93.9** | **87.9** | **97.6** | 98.9 | **534.3** | **37.6** | **62.7** | **73.9** | **56.7** | **80.8** | **88.7** | **400.4** |
| 0.2 | 66.9 | 88.5 | 93.6 | 86.4 | 97.1 | 99.0 | 531.5 | 36.9 | 62.2 | 73.4 | 55.6 | 80.5 | 88.3 | 396.9 |

## 5 CONCLUSION

This work has studied a practically important but seldom investigated problem as unpaired image-text matching. To deal with this problem, we propose multimodal aligned semantic knowledge, which leverages word embeddings as bridges to associate words with prototypes, capturing semantic relationships between words, and further utilizing information from OOD words. Additionally, the introduction of prototype consistency contrastive loss effectively mitigates the impact of variance in unpaired matching. Code is available at https://github.com/AndroidDevelopersTools/MASK.

ACKNOWLEDGMENTS

This work was supported by the National Natural Science Foundation of China (62376138, 92367202), and in part by the Jinan-NTU Green Technology Research Institute (GreenTRI).

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

# A  TABLE OF NOTATION

We list the notation used in this paper in Table 7, for the convenience of reference.

Table 7: Notation used in the paper.

| Symbol | Description |
|---|---|
| $w_k$ | Word embedding directly from pre-trained word vectors |
| $v_k$ | Learnable prototypical region representation (prototype) of $k$-th semantic concept |
| $r_j$ | Raw region representation from pre-trained object detection model Faster-RCNN |
| $\mu_j$ | The mean of the latent space corresponds to the region representation |
| $\sigma_j$ | The variance of the latent space |
| $z$ | Random vector sampled from Gaussian distribution |
| $R$ | $\{r_n\}_{n=1,...,B}$ $(R \in \mathcal{R}^{B \times M})$ |
| $\mu_+$ | Region representation associated with the prototypical region representations $v_k$ |
| $V$ | $\{w_k\}_{k=1,...,B}$ $(V \in \mathcal{R}^{B \times N})$ |
| $d_s$ | Distance metrics, e.g., cosine similarity |
| $U$ | $\{v_j\}_{j=1,...,J}$ $(U \in \mathcal{R}^{J \times Z})$ |
| $\rho(\cdot)$ | Max-mean pooling |
| $w_{out}$ | Word embedding $w_{out}$ corresponding to OOD word |
| $s_q$ | Similarity score |
| $v_{out}$ | Prototypical region representation $v_{out}$ corresponding to word embedding $w_{out}$ |
| $\tilde{s}^k$ | Top-$k$ similarity vector from existing multimodal model |
| $s^k$ | Top-$k$ similarity vector from MASK |

# B  PROOF: EQ. 8 FACILITATES THE CONSTRUCTION OF CORRESPONDING PROTOTYPES FOR OOD WORDS.

Given a paired multimodal knowledge $\{(w_k, v_k)\}_{k=1,...,K}$, where $w_k \in \mathcal{R}^N$ and $v_k \in \mathcal{R}^M$ are the word embedding and prototypical region representation of the $k$-th semantic concept, respectively, and $K$ is the total number of semantic concepts. We sample from the multimodal knowledge $\{(w_k, v_k)\}_{k=1,...,K}$ to obtain a subset $\{(w_k, v_k)\}_{k=1,...,\hat{K}}$. Our objective is to utilize the known word embedding $w_{out}$ and $\{(w_k, v_k)\}_{k=1,...,\hat{K}}$ to construct $v_{out}$ such that the following equation is satisfied:

$$d_s(f(v_k),\ f(v_{out})) \propto d_s(v_k,\ v_{out}),\ \ k \in \{1, \ldots, \hat{K}\}, \tag{16}$$

$$and\ \ f(v_{out}) \approx w_{out}. \tag{17}$$

We first rewrite Eq. 16 into an equidistant form. Then, there exists a $\beta > 0$ such that:

$$d_s(f(v_k),\ f(v_{out})) = \beta\, d_s(v_k,\ v_{out}). \tag{18}$$

Define the scaled mapping $F := \frac{1}{\beta}\, f$. For any $k, out$:

$$d_s(F(v_k),\ F(v_{out})) = d_s(v_k,\ v_{out}). \tag{19}$$

That is, $F$ precisely preserves the euclidean distance among these prototypes. An isometric mapping in euclidean space exhibits a well-defined structural property, which is a fundamental result in classical geometry and functional analysis. Specifically, any distance-preserving mapping defined on the euclidean space must be a rigid transformation (Iovino, 2021). In other words, there exists an orthogonal matrix $A$ that represents a rotation or reflection, and a translation vector $t$, such that the mapping can be expressed as:

$$F(v_k) = A v_k + t. \tag{20}$$

Therefore, the $f$ can be expressed as:

$$f(v_k) = \beta A v_k + \beta t. \tag{21}$$

Since $A$ is an orthogonal matrix, its inverse $A^{-1}$ must exist and is equal to its transpose $A^{\top}$. Therefore, we can derive:

$$v_k = A^{\top}(F(v_k) - t) = A^{\top}(\frac{1}{\beta} f(v_k) - t). \tag{22}$$

For any given $w_k$ in the knowledge, according to Eq. 17, we have:

$$v_k = A^{\top}(\frac{1}{\beta} w_k - t). \tag{23}$$

Therefore, we only need to determine the values of $A$, $t$, and $\beta$ to obtain the visual prototype vector $v_{out}$ corresponding to the OOD words $w_{out}$. Extending Eq. 20 to the general case where $A$ is no longer a strictly orthogonal matrix, we obtain the following equation using the column orthogonality condition:

$$||Av_k - Av_{out}||_2 = ||v_k - v_{out}||_2 \Leftrightarrow A^{\top}A = I_{min\{M,N\}}, \tag{24}$$

Here, there exists an $A^{\top}$ such that Eq. 23 holds. We do not need to know the values of $A$, $t$, and $\beta$ a priori, as they can be uniquely determined from the paired knowledge $\{(w_k, v_k)\}_{k=1,...,K}$ using standard similarity Procrustes decomposition. First, we construct matrix $W_1 = [v_1, \ldots, v_K]$ and matrix $W_2 = [w_1, \ldots, w_K]$. Next, we compute the centroids of each set:

$$\bar{v} = \frac{1}{K}\sum_{k=1}^{K} v_k \in \mathcal{R}^M, \quad \bar{w} = \frac{1}{K}\sum_{k=1}^{K} w_k \in \mathcal{R}^N. \tag{25}$$

We centralize the knowledge using the computed centroids $\bar{v}$ and $\bar{w}$:

$$\tilde{W}_1 = [v_1 - \bar{v}, \ldots, v_K - \bar{v}] \in \mathcal{R}^{M \times K}, \quad \tilde{W}_2 = [w_1 - \bar{w}, \ldots, w_K - \bar{w}] \in \mathcal{R}^{N \times K}. \tag{26}$$

From here, we can separate out $t$, and subsequently only need to consider the linear transformation and scaling. Next, we construct the covariance matrix $C$:

$$C = \tilde{W}_1 \tilde{W}_2^{\top} \in \mathcal{R}^{M \times N}. \tag{27}$$

Let $M > N$. And then, we utilize SVD decomposition to obtain the values of $A$, $t$, and $\beta$:

$$C = U\Sigma V^{\top}, \quad U \in \mathcal{R}^{M \times M}, \Sigma \in \mathcal{R}^{M \times N}, V^{\top} \in \mathcal{R}^{N \times N}, \tag{28}$$

$$A = U \begin{bmatrix} I_N \\ 0_{(M-N) \times N} \end{bmatrix} V^{\top}, \quad A \in \mathcal{R}^{M \times N}, \tag{29}$$

$$\beta = \frac{trace(A\tilde{W}_2\tilde{W}_1^{\top})}{trace(\tilde{W}_2\tilde{W}_1^{\top})}, \tag{30}$$

$$t = \bar{w} - \beta A\bar{v}, \quad t \in \mathcal{R}^M. \tag{31}$$

Finally, we substitute the values of $A$, $t$, and $\beta$ into Eq. 23 to obtain the prototype vector $v_{out}$ corresponding to the word embedding $w_{out}$.

## C   PROOF: THE COSINE SIMILARITY IN EQ. 9 IS A REASONABLE DISTANCE METRIC $d_s$.

Euclidean space possesses strict linear structure preservation properties. It is the only metric satisfying translation and rotation invariance, and is naturally compatible with linear mappings. This allows the structural alignment problem between visual and linguistic spaces to be transformed into a standard orthogonal Procrustes problem. Therefore, we can equate Eq. 19 with Eq. 20 in euclidean space.

Cosine similarity and euclidean distance are not inherently equivalent. However, after vector normalization, there exists a strict monotonic mapping between cosine similarity and euclidean distance. This implies that, cosine similarity can be regarded as a form of "Euclidean-like distance," thereby satisfying the prerequisites for similarity transformations. Let two vectors $w_k$ and $w'_k$, their cosine similarity is defined as:

$$cos(w_k, w'_k) = \frac{w_k \cdot w'_k}{||w_k|| \, ||w'_k||}. \tag{32}$$

where $|| \cdot ||$ denotes the $L1$-norm. The euclidean distance is defined as:

$$d_E(w_k, w'_k) = ||w_k - w'_k||. \tag{33}$$

We normalize each vector such that $||w_k||=||w'_k||=1$, yielding:

$$d_E(w_k, w'_k)^2 = ||w_k - w'_k||^2 = ||w_k||^2 + ||w'_k||^2 - 2 \cdot w_k w'_k = 2 - 2cos(w_k, w'_k). \tag{34}$$

$$cos(w_k, w'_k) = 1 - \frac{1}{2}d_E(w_k, w'_k)^2. \tag{35}$$

Combining Eq. 34 and Eq. 35, it can be concluded that after normalization, a one-to-one monotonic functional relationship exists between cosine similarity and euclidean distance.

$$\mathcal{L}_{cm} = \underbrace{\mathbb{E}\left[(1 - cos(\frac{w_i}{\|w_i\|_2}, \frac{w'_i}{\|w'_i\|_2}))\right]}_{word-alignment} + \underbrace{\mathbb{E}\left[((cos(\frac{w'_i}{\|w'_i\|_2}, \frac{w'_j}{\|w'_j\|_2}) - cos(\frac{\mu_i}{\|\mu_i\|_2}, \frac{\mu_j}{\|\mu_j\|_2})))^2\right]}_{structure-preserving}. \tag{36}$$

The loss $\mathcal{L}_{cm}$ comprises two components: the first item enforces alignment between predicted and pre-trained word embeddings, while the second ensures that region representations capture the structural relationships among words.

$$\frac{\partial(1 - cos(\frac{w_i}{\|w_i\|_2}, \frac{w'_i}{\|w'_i\|_2}))}{\partial \frac{w'_i}{\|w'_i\|_2}} = \frac{(w'_i{}^\top w_i)w'_i}{\|w'_i\|_2^2 \|w_i\|_2} - \frac{w_i}{\|w_i\|_2}. \tag{37}$$

$$\frac{\partial(1 - cos(\frac{w_i}{\|w_i\|_2}, \frac{w'_i}{\|w'_i\|_2}))}{\partial \frac{w'_i}{\|w'_i\|_2}} = 0 \implies w_i = w'_i. \tag{38}$$

$$\frac{\partial([(cos(\frac{w'_i}{\|w'_i\|_2}, \frac{w'_j}{\|w'_j\|_2}) - cos(\frac{\mu_i}{\|\mu_i\|_2}, \frac{\mu_j}{\|\mu_j\|_2}))^2])}{\partial \frac{w'_i}{\|w'_i\|_2}}$$
$$= 2(cos(\frac{w'_i}{\|w'_i\|_2}, \frac{w'_j}{\|w'_j\|_2}) - cos(\frac{\mu_i}{\|\mu_i\|_2}, \frac{\mu_j}{\|\mu_j\|_2}))\frac{\partial(cos(\frac{w'_i}{\|w'_i\|_2}, \frac{w'_j}{\|w'_j\|_2}))}{\partial \frac{w'_i}{\|w'_i\|_2}}. \tag{39}$$

$$\frac{\partial(cos(\frac{w'_i}{\|w'_i\|_2}, \frac{w'_j}{\|w'_j\|_2}))}{\partial \frac{w'_i}{\|w'_i\|_2}} = \frac{w'_j}{\|w'_j\|_2} - \frac{(w'_i \cdot w'_j)w'_i}{\|w'_i\|_2^2 \|w'_j\|_2}. \tag{40}$$

$$\frac{\partial([(cos(\frac{w'_i}{\|w'_i\|_2}, \frac{w'_j}{\|w'_j\|_2}) - cos(\frac{\mu_i}{\|\mu_i\|_2}, \frac{\mu_j}{\|\mu_j\|_2}))^2])}{\partial \frac{w'_i}{\|w'_i\|_2}} = 0$$
$$\implies cos(\frac{w'_i}{\|w'_i\|_2}, \frac{w'_j}{\|w'_j\|_2}) = cos(\frac{\mu'_i}{\|\mu'_i\|_2}, \frac{\mu'_j}{\|\mu'_j\|_2}). \tag{41}$$

## D  PROOF: THE OBJECTIVE EXISTENCE OF THE ISOMETRIC HYPOTHESIS.

Proof: For any finite set of region representations $\{\mu_i\}_{i=1}^n$, there necessarily exists a mapping $f$ in an appropriate Euclidean space such that:

$$d_s(f(\mu_i; \Theta_f), f(\mu_j; \Theta_f)) \propto d_s(\mu_i, \mu_j). \tag{42}$$

Given $n$ non-zero vectors $\mu_1, \ldots, \mu_n \in \mathcal{R}^Z$, normalize them to obtain:

$$\hat{\mu}_i := \frac{\mu_i}{\|\mu_i\|_2}, \quad i = 1, \ldots, n. \tag{43}$$

We aim to construct a set of vectors $\{y_i\}_{i=1}^n$ in a Euclidean space as $f(\mu_i)$, and demonstrate that $cos(y_i, y_j) = cos(\mu_i, \mu_j)$ can be achieved. Next, we organize the pairwise similarities into a matrix, facilitating the use of spectral decomposition to construct vectors that satisfy the given inner product relationships. We define the $n \times n$ Gram matrix $G$ as follows:

$$G_{ij} := cos(\hat{\mu}_i, \hat{\mu}_j) = \hat{\mu}_i^\top \hat{\mu}_j. \tag{44}$$

Here $G$ is a real symmetric positive semi-definite (PSD) matrix, so for any vector $z \in \mathcal{R}^n$, it satisfies:

$$z^\top G z = \|\sum_i z_i \hat{\mu}_i\|_2^2 \geq 0, \tag{45}$$

where the $G_{ii} = 1$. According to the properties of real symmetric PSD matrix, there exist an orthogonal matrix $\mathcal{U} \in \mathcal{R}^{n \times n}$ and a diagonal matrix $\Lambda = diag(\lambda_1, \ldots \lambda_n)(\lambda_k \geq 0)$ such that:

$$G = \mathcal{U} \Lambda \mathcal{U}^\top, \tag{46}$$

where $r = rank(G) \leq n$. Take the non-zero eigenvalue part of $\Lambda$ as $\Lambda_r \in \mathcal{R}^{r \times r}$, and take the first $r$ columns of the corresponding eigenvectors to obtain $\mathcal{U}_r \in \mathcal{R}^{n \times r}$:

$$G = \mathcal{U}_r \Lambda_r \mathcal{U}_r^\top, \tag{47}$$

$$Y := \mathcal{U}_r \Lambda_r^{1/2} \in \mathcal{R}^{n \times r}, \tag{48}$$

where we write $Y$ in row-vector form, and denote its $i$-th row as $y_i^\top$ ($y_i \in \mathcal{R}^r$):

$$(YY^\top)_{ij} = y_i^\top y_j = (\mathcal{U}_r \Lambda_r^{1/2})(\mathcal{U}_r \Lambda_r^{1/2})^\top {}_{ij} = \mathcal{U}_r \Lambda_r \mathcal{U}_r^\top {}_{ij} = G_{ij}. \tag{49}$$

Therefore, we have constructed $n$ vectors $y_1, \ldots, y_n \in \mathcal{R}^r$ that satisfy the inner-product relation $y_i^\top y_j = G_{ij}$:

$$\|y_i\|_2^2 = y_i^\top y_i = G_{ii} = 1, \tag{50}$$

$$cos(y_i, y_j) = \frac{y_i^\top y_j}{\|y_i\| \ \|y_j\|} = \frac{G_{ij}}{1 \cdot 1} = cos(\hat{\mu}_i, \hat{\mu}_j), \tag{51}$$

$$cos(y_i, y_j) = cos(\hat{\mu}_i, \hat{\mu}_j) = cos(\mu_i, \mu_j). \tag{52}$$

By setting $f(\mu_i) := y_i$ $(i = 1, \ldots, n)$, we obtain:

$$cos(f(\mu_i), f(\mu_j)) = cos(\mu_i, \mu_j) \ \forall \ i, j, \tag{53}$$

$$d_s(f(\mu_i; \Theta_f), \ f(\mu_j; \Theta_f)) \propto d_s(\mu_i, \ \mu_j). \tag{54}$$

## E   PROOF: $\mathcal{L}_{cm}$ (EQ. (9)) ENCOURAGES $f$ TO APPROACH THE ISOMETRY ASSUMPTION (EQ. (8)).

Given the pre-trained word embeddings $w_i$ and the predicted word embeddings $w_i' = f(\mu_i; \Theta_f)$, we normalize them to obtain:

$$\hat{w}_i' := \frac{w_i'}{\|w_i'\|}, \quad \hat{w}_i := \frac{w_i}{\|w_i\|}. \tag{55}$$

Similarly, we normalize the region representations to obtain $\hat{\mu}_i := \frac{\mu_i}{\|\mu_i\|}$. $\mathcal{L}_{cm}$ consists of $\mathcal{L}_{word}$ and $\mathcal{L}_{struct}$ (Appendix F). For a given batch size $B$, there exist constants $\epsilon_w > 0$ and $\epsilon_s > 0$ such that, when $\mathcal{L}_{word} \leq \epsilon_w$ and $\mathcal{L}_{struct} \leq \epsilon_s$, the following inequality holds:

$$\left| cos(\hat{w}_i', \hat{w}_j') - cos(\mu_i, \mu_j) \right| \leq 4\sqrt{2\epsilon_w} + \sqrt{\frac{\epsilon_s}{M}}, \tag{56}$$

where $M$ denotes the number of ordered pairs ($i \neq j$), and $i, j \in B$. When $\epsilon_w \to 0$ and $\epsilon_s \to 0$, $cos(\hat{w}_i', \hat{w}_j') - cos(\mu_i, \mu_j)$ admits an upper bound arbitrarily close to 0, meaning that the cosine similarity between the mapped vector pairs approaches that in the original $\mu$-space, thereby achieving $d_s(f(\mu_i; \Theta_f), \ f(\mu_j; \Theta_f)) \propto d_s(\mu_i, \ \mu_j)$.

We first convert the point-wise cosine error into the Euclidean difference between vectors. According to Appendix C, we have:

$$\|\hat{w}'_i - \hat{w}_i\|_2^2 = 2(1 - cos(\hat{w}'_i, \hat{w}_i)). \tag{57}$$

For each case where $i \in B$, let $\delta_i := 1 - cos(\hat{w}'_i, \hat{w}_i) \geq 0$. Then:

$$\|\hat{w}'_i - \hat{w}_i\|_2 = \sqrt{2\delta_i}, \tag{58}$$

$$\sum_{i \in B} \delta_i = \mathcal{L}_{word} \leq \epsilon_w, \tag{59}$$

$$\delta_i \leq \epsilon_w \Rightarrow \quad \|\hat{w}'_i - \hat{w}_i\|_2 \leq \sqrt{2\epsilon_w}. \tag{60}$$

To quantify how point-wise alignment errors influence the pairwise cosine discrepancies, we characterize the variation of individual entries in the cosine similarity matrix using the norm difference. Then:

$$\left|cos(\hat{w}'_i, \hat{w}'_j) - cos(\hat{w}_i, \hat{w}_j)\right| = \left|\hat{w}'_i \cdot \hat{w}'_j - \hat{w}_i \cdot \hat{w}_j\right|, \tag{61}$$

$$
\begin{aligned}
&\left|\hat{w}'_i \cdot \hat{w}'_j - \hat{w}_i \cdot \hat{w}_j\right| \\
&= \left|(\hat{w}'_i - \hat{w}_i) \cdot \hat{w}'_j + \hat{w}_i \cdot (\hat{w}'_j - \hat{w}_j)\right| \\
&\leq \|\hat{w}'_i - \hat{w}_i\|_2 \cdot \|\hat{w}'_j\|_2 + \|\hat{w}_i\|_2 \cdot \|\hat{w}'_j - \hat{w}_j\|_2 \\
&\leq \|\hat{w}'_i - \hat{w}_i\|_2 + \|\hat{w}'_j - \hat{w}_j\|_2
\end{aligned}
\tag{62}
$$

$$\left|\hat{w}'_i \cdot \hat{w}'_j - \hat{w}_i \cdot \hat{w}_j\right| \leq \sqrt{2\epsilon_w} + \sqrt{2\epsilon_w} = 2\sqrt{2\epsilon_w}. \tag{63}$$

To quantify the influence of the structural loss $\mathcal{L}_{struct}$ on the reconstruction error, we convert the structural loss into an upper bound on the per-pair error. Let the per-pair error be defined as $e_{ij} := cos(\hat{w}'_i, \hat{w}'_j) - cos(\mu_i, \mu_j)$. Then we obtain:

$$\sum_{i \neq j} e_{ij}^2 = \mathcal{L}_{struct} \leq \epsilon_s, \tag{64}$$

$$\max_{i \neq j} |e_{ij}| \leq \sqrt{\sum_{i \neq j} e_{ij}^2} \leq \sqrt{\epsilon_s}. \tag{65}$$

If we distribute $\epsilon_s$ uniformly across all pairs, then the absolute error of each pair is bounded above by $\epsilon_s/M$. Then:

$$
\begin{aligned}
&\left|cos(\hat{w}'_i, \hat{w}'_j) - cos(\mu_i, \mu_j)\right| \\
&= \left|cos(\hat{w}'_i, \hat{w}'_j) - cos(\hat{w}_i, \hat{w}_j) + cos(\hat{w}_i, \hat{w}_j) - cos(\mu_i, \mu_j)\right| \\
&\leq \left|cos(\hat{w}'_i, \hat{w}'_j) - cos(\hat{w}_i, \hat{w}_j)\right| + \left|cos(\hat{w}_i, \hat{w}_j) - cos(\mu_i, \mu_j)\right| \\
&\leq 2\sqrt{2\epsilon_w} + \left|cos(\hat{w}_i, \hat{w}_j) - cos(\hat{w}'_i, \hat{w}'_j) + cos(\hat{w}'_i, \hat{w}'_j) - cos(\mu_i, \mu_j)\right| \\
&\leq 4\sqrt{2\epsilon_w} + \sqrt{\frac{\epsilon_s}{M}}
\end{aligned}
\tag{66}
$$

Therefore, by enforcing $\epsilon_w \to 0$ and $\epsilon_s \to 0$ during training, we obtain the following expression:

$$\left|cos(\hat{w}'_i, \hat{w}'_j) - cos(\mu_i, \mu_j)\right| \to 0. \tag{67}$$

$$d_s(f(\mu_i; \Theta_f), f(\mu_j; \Theta_f)) \propto d_s(\mu_i, \mu_j). \tag{68}$$

# F    PROOF: THE $w'$ IN EQ. (9) CANNOT BE REPLACED BY $w$.

The $w'$ in Eq. (9) cannot be replaced by $w$. Doing so would remove the "structure-preservation" supervision imposed on the $MTM$ model $f$, as the second term in Eq. (9) would no longer contribute a meaningful loss for training $f$. The loss function recommended in the article for preserving structure (the second term of Eq.(9)) is defined as $\mathcal{L}_{struct}^{A}$:

$$\mathcal{L}_{struct}^{A} = \mathbb{E}[(cos(\frac{w_i'}{\|w_i'\|_2}, \frac{w_j'}{\|w_j'\|_2}) - cos(\frac{\mu_i}{\|\mu_i\|_2}, \frac{\mu_j}{\|\mu_j\|_2}))^2]. \tag{69}$$

The loss function obtained after replacing $w'$ with $w$ in $\mathcal{L}_{struct}^{A}$ is defined as $\mathcal{L}_{struct}^{B}$:

$$\mathcal{L}_{struct}^{B} = \mathbb{E}[(cos(\frac{w_i}{\|w_i\|_2}, \frac{w_j}{\|w_j\|_2}) - cos(\frac{\mu_i}{\|\mu_i\|_2}, \frac{\mu_j}{\|\mu_j\|_2}))^2]. \tag{70}$$

Next, we compare the constraint capabilities of $\mathcal{L}_{struct}^{A}$ and $\mathcal{L}_{struct}^{B}$ on the $MTM$ model $f$ (with parameters $\Theta_f$).

## F.1    FROM THE PERSPECTIVE OF THE GRADIENT OF THE LOSS FUNCTION

For the loss function $\mathcal{L}_{struct}^{A}$, we apply the chain rule to $\Theta_f$ as follows:

$$\frac{\partial \mathcal{L}_{struct}^{A}}{\partial \Theta_f} = 2\mathbb{E}(cos(\frac{w_i'}{\|w_i'\|_2}, \frac{w_j'}{\|w_j'\|_2}) - cos(\frac{\mu_i}{\|\mu_i\|_2}, \frac{\mu_j}{\|\mu_j\|_2})) \cdot \frac{\partial\ cos(\frac{w_i'}{\|w_i'\|_2}, \frac{w_j'}{\|w_j'\|_2})}{\partial \Theta_f}. \tag{71}$$

$$\frac{\partial\ cos(\frac{w_i'}{\|w_i'\|_2}, \frac{w_j'}{\|w_j'\|_2})}{\partial \Theta_f} = \frac{\partial\ cos(\frac{w_i'}{\|w_i'\|_2}, \frac{w_j'}{\|w_j'\|_2})}{\partial w_i'} \cdot \frac{\partial w_i'}{\partial \Theta} + \frac{\partial\ cos(\frac{w_i'}{\|w_i'\|_2}, \frac{w_j'}{\|w_j'\|_2})}{\partial w_j'} \cdot \frac{\partial w_j'}{\partial \Theta}. \tag{72}$$

As long as the parameters of $f$ influence $w_i'$ (i.e., $\frac{\partial w_i'}{\partial \Theta} \neq 0$), the above equation generally does not vanish. Consequently, $\mathcal{L}_{struct}^{A}$ produces a nonzero gradient that drives the update of $\Theta_f$, forcing the mapping $f$ to bring $cos(\frac{w_i'}{\|w_i'\|_2}, \frac{w_j'}{\|w_j'\|_2})$ closer to $cos(\frac{\mu_i}{\|\mu_i\|_2}, \frac{\mu_j}{\|\mu_j\|_2})$. Therefore, $\mathcal{L}_{struct}^{A}$ directly imposes constraints on the $MTM$ model $f$.

Moreover, the $cos(\frac{w_i}{\|w_i\|_2}, \frac{w_j}{\|w_j\|_2})$ in $\mathcal{L}_{struct}^{B}$ is a constant. We apply the chain rule to $\Theta_f$ as follows:

$$\mathcal{L}_{struct}^{B} = \mathbb{E}[(\underbrace{cos(\frac{w_i}{\|w_i\|_2}, \frac{w_j}{\|w_j\|_2})}_{constant} - cos(\frac{\mu_i}{\|\mu_i\|_2}, \frac{\mu_j}{\|\mu_j\|_2}))^2]. \tag{73}$$

$$\frac{\partial\ \mathcal{L}_{struct}^{B}}{\partial \Theta_f} = 0. \tag{74}$$

$\mathcal{L}_{struct}^{B}$ provides no constraints or gradient information on $f$, and therefore cannot compel the mapping $f$ to preserve input similarity in any manner. Although $\mu$ is affected, this influence does not propagate to $\Theta_f$, making it impossible to fulfill the objective of structural preservation.

## F.2    FROM THE PERSPECTIVE OF THE JOINT OPTIMIZATION OBJECTIVE

The loss function recommended in the article for word alignment (the first term of Eq.(9)) is defined as $\mathcal{L}_{word}$:

$$\mathcal{L}_{word} = \mathbb{E}[(1 - cos(\frac{w_i}{\|w_i\|_2}, \frac{w_i'}{\|w_i'\|_2}))]. \tag{75}$$

Next, we compared the differences between the two loss functions when jointly optimized with $\mathcal{L}_{word}$. The joint loss function for $\mathcal{L}_{word}$ and $\mathcal{L}_{struct}^{A}$ is expressed as follows:

$$\mathcal{L}_{cm}^{A} = \mathcal{L}_{word} + \mathcal{L}_{struct}^{A}. \tag{76}$$

Both loss terms involve $\Theta_f$ (through $w_i'$) and therefore jointly constrain the mapping. Specifically, $\mathcal{L}_{word}$ pulls each single-point mapping $w_i'$ toward its corresponding $w_i$, while $\mathcal{L}_{struct}^{A}$ enforces that $f$

preserve the pairwise similarity structure among samples, consistent with the semantic relationships in the $\mu$-space (prototype). The joint effect of these two losses ensures that the learned mapping $f$ achieves both accurate pointwise alignment and global structural preservation.

The joint loss function for $\mathcal{L}_{word}$ and $\mathcal{L}_{struct}^B$ is expressed as follows:

$$\mathcal{L}_{cm}^B = \mathcal{L}_{word} + \mathcal{L}_{struct}^B. \tag{77}$$

It is worth noting that $\mathcal{L}_{struct}^B$ does not impose any constraints on the model $f$. Consequently, $\mathcal{L}_{word}$ is the only term that directly supervises $f$, and minimizing it merely pulls $w_i'$ toward its corresponding $w_i$. In contrast, $\mathcal{L}_{struct}^B$ adjusts the $\mu$-space (governed by the parameters of the $PAE$ model) to make $cos(\frac{\mu_i}{\|\mu_i\|_2}, \frac{\mu_j}{\|\mu_j\|_2})$ approach $cos(\frac{w_i}{\|w_i\|_2}, \frac{w_j}{\|w_j\|_2})$, but it provides no mechanism to enforce structural preservation in the mapping produced by $f$.

## G   PROOF: THE PROTOTYPE CONSISTENCY CONTRASTIVE LEARNING LOSS $\mathcal{L}_{cl}$ ENHANCES THE DISCRIMINABILITY OF REGION REPRESENTATIONS.

The prototype consistency contrastive learning loss enhances the discriminability of region representations by reducing intra-word variance and increasing inter-word separation. Intra-word variance $\sigma_k^2$ and inter-word separation $D_{k,k'}$ can be defined as follows:

$$\sigma_k^2 = \frac{1}{J_k} \sum_{j=1}^{J_k} ||\mu_j - v_k||^2, \tag{78}$$

$$D_{k,k'} := ||v_k - v_{k'}||^2. \tag{79}$$

For any $\mu_j$:

$$||\mu_j - v_k||^2 = ||\mu_j - \bar{\mu} + \bar{\mu} - v_k||^2 = ||\mu_j - \bar{\mu}||^2 + ||\bar{\mu} - v_k||^2 + 2\langle \mu_j - \bar{\mu}, \bar{\mu} - v_k \rangle, \tag{80}$$

$$\sigma_k^2 \approx \frac{1}{J_k} \sum_{j=1}^{J_k} ||\mu_j - \bar{\mu}||^2 + ||\bar{\mu} - v_k||^2, \tag{81}$$

where $\bar{\mu}$ serves as the temporary mean for a given batch. The intra-word variance can be decomposed into the within-batch intra-word variance and the deviation between the batch-specific word centers and the prototypes. The loss $\mathcal{L}_{cl}$ is updated along the gradient direction:

$$\mu_j \longleftarrow \mu_j - \eta(\mu_j - v_k). \tag{82}$$

$$\mu_j(t+1) = (1-\eta)\,\mu_j(t) + \eta v_k. \tag{83}$$

$$\sigma_k^2(t+1) = (1-\eta)^2 \sigma_k^2(t). \tag{84}$$

As iterations progress, $\sigma_k^2$ gradually approaches zero, causing samples within the same word to cluster tightly around their prototypes and thereby enhancing the discriminability of the region representations. The loss $\mathcal{L}_{cl}$ is updated by moving along the gradient direction while simultaneously being repelled away from other prototypes:

$$\mu_j \longleftarrow \mu_j + \eta \sum_{k \neq k'} (\mu_j - v_{k'}). \tag{85}$$

$$v_k' = \frac{1}{J_k} \sum_{j=1}^{J_k} \mu_j(t+1) = v_k + \eta \sum_{k \neq k'} (v_k - v_{k'}). \tag{86}$$

$$D_{k,k'}(t+1) = ||v_k' - v_{k'}'||^2 = ||v_k - v_{k'} + \eta(\sum_{k \neq \hat{k}}(v_k - v_{\hat{k}}) - \sum_{k' \neq \hat{k}}(v_{k'} - v_{\hat{k}})||^2 \tag{87}$$

At convergence, each prototype is positioned as far as possible from all others, resulting in distinct separation between different prototypes and further enhancing the discriminability of the region representations.

## H  FINE-TUNED DOMAIN KNOWLEDGE

To better adapt the multimodal aligned semantic knowledge to specific datasets, we can further fine-tune the prototypical region representations to obtain domain-specific knowledge. This fine-tuning step is optional and depends on the availability of unpaired data in the target dataset. Notably, the multimodal aligned semantic knowledge alone can also be applied directly and achieves strong performance. Given that annotating paired image-text data is costly, while unpaired images and texts are typically more accessible, we adopt a bidirectional region-word cycle-consistent learning approach in an unpaired learning setting.

In particular, given a batch of unpaired images and texts, we first obtain a set of raw region representations $R = \{r_n\}_{n=1,...,I} (R \in \mathcal{R}^{I \times M})$ using the pre-trained Faster-RCNN above and a set of parsed words through tokenization operation implemented via NLTK. Then, we use the knowledge as a cross-modal bridge to represent all the words into the corresponding prototypical region representations $u = \{v_j\}_{j=1,...,J} (u \in \mathcal{R}^{J \times Z})$. For the set of regions $R$, we extract region representations $\mu \in \mathcal{R}^{I \times Z}$ by utilizing the $PAE$ model $h$.

We utilize a bidirectional region-word cycle-consistent loss to learn a parametric transformation matrix $W \in \mathcal{R}^{Z \times Z}$ [3]. This loss incorporates two cross-modal similarity measurement processes: region-to-word (R2W) and word-to-region (W2R). In the R2W process, similarities between each word and all regions are first computed, and these similarities are then used as weights to aggregate all regions into a reconstructed word representation. Conversely, in the W2R process, similarities between each region and all words are computed to reconstruct regions from the word representations. The corresponding formulations are

$$S = uW(\mu W)^\top, \tag{88}$$

$$\hat{u} = softmax(S)\, \mu W, \; \hat{\mu} = softmax(S^\top)\, uW, \tag{89}$$

where $S \in \mathcal{R}^{J \times I}$ is the similarity matrix between transformed word and region representations, $\hat{u} \in \mathcal{R}^{J \times Z}$ contains the reconstructed word representations from region representations, and $\hat{\mu} \in \mathcal{R}^{I \times Z}$ contains the reconstructed region representations from word representations.

Each original word (or region) representation is then compared with its reconstructed counterpart to determine whether they correspond to the same entity. By minimizing the cross-entropy between the predicted and ground-truth labels, we obtain a self-supervised loss, $\mathcal{L}_{ss}$, which is used to optimize $W$:

$$\hat{Y}^{R2W} = softmax(uW\hat{u}^\top)^\top, \; \hat{Y}^{W2R} = softmax(\mu W\hat{\mu}^\top)^\top, \tag{90}$$

$$\mathcal{L}^{R2W} = -\sum_{j=1}^{J} y_j^\top log(\hat{y}_j), \tag{91}$$

$$\mathcal{L}^{W2R} = -\sum_{n=1}^{I} y_n^\top log(\hat{y}_n), \tag{92}$$

$$\mathcal{L}^{ss} = \mathcal{L}^{R2W} + \mathcal{L}^{W2R}, \tag{93}$$

where $\hat{Y}^{R2W} \in \mathcal{R}^{J \times J}$ and $\hat{Y}^{W2R} \in \mathcal{R}^{I \times I}$ are two matrices including the predicted labels in R2W and W2R directions, respectively. In $\hat{Y}^{R2W}$ and $\hat{Y}^{W2R}$, the $j$-th and $n$-th columns are denoted as $\hat{y}_j$ and $\hat{y}_n$, respectively. $y_j$ and $y_n$ are two groundtruth label vectors, whose the $j$-th and $n$-th values are ones and the rest are zeros. After the cycle consistent learning, we can use the learnable $W$ to transform all prototypical region representations into the fine-tuned domain knowledge, denoted as $\{(w_k, \hat{v}_k)\}_{k=1,...,K}$, where $\hat{v}_k = v_k W \in \mathcal{R}^Z$.

## I  EXPERIMENTAL SETTING

### I.1  DATASET

The details of experimental datasets and metrics are described as follows.

---

[3] We also stack multiple transformation matrices in a nonlinear way, but find it does not necessarily lead to better performance

**Flickr30K** (Young et al., 2014) consists of 31,783 images collected from the Flickr website. Each image has 5 human annotated texts. We use the public training, validation and testing splits, which contain 29,783, 1,000 and 1,000 images, respectively.

**MSCOCO** (Lin et al., 2014) consists of 123,287 images, each of which is associated with 5 texts. We use the public training, validation and testing splits, with 113,287, 5,000 and 5,000 images, respectively.

## I.2 IMPLEMENTATION DETAILS

In the multimodal aligned semantic knowledge, we collect all words from the VG dataset and filter out some special characters and rare words, resulting in a total of $K$=12,385 semantic concepts. For each image, we initially employ the pre-trained object detection model Faster-RCNN [4] to extract raw region representations, setting the number of detected regions to $I$=36 and the dimensionality of each region representation to $M$=2048. For each word, we obtain its word embedding using the pre-trained word vectors glove-840B-300d [5]. The batch size is 4096 for the first 200 epochs and 2048 for the next 200 epochs. The trade-off factors $\lambda_1$ and $\lambda_2$ are set to 3. The sampling size $m$ is set to 10. We use the Adam to optimize the loss with a learning rate of 1e-4.

## I.3 PRETRAINED MODELS AND WORD VECTORS

**Faster-RCNN** is a widely used deep learning model for object detection, tasked with both identifying and localizing objects within an image. Building on earlier approaches such as R-CNN and Fast R-CNN, it introduces a Region Proposal Network (RPN) that generates object proposals directly within the model. This integration significantly improves both speed and accuracy, enabling efficient, real-time detection of multiple objects with high precision. We use Detectron2 as the backend to support comprehensive functions, including training, testing, and feature extraction. Additionally, we migrate the pre-trained Caffe-based model from the original repository, ensuring that it extracts visual features consistent with the original model, with deviations of less than 0.01.

**GloVe** is an unsupervised learning algorithm designed to generate vector representations of words. It is trained on aggregated global word-word co-occurrence statistics from a corpus, producing embeddings that capture meaningful semantic relationships and exhibit interpretable linear substructures within the word vector space. We obtain word embedding using the pre-trained word vectors glove-840B-300d. It is a set of pre-trained word embeddings derived from the GloVe model developed by Stanford University, trained specifically on Common Crawl data. The model is built using approximately 840B tokens, resulting in a vocabulary of 2.2 million words and producing 300-dimensional word vectors.

**CLIP** [6] is a cross-modal pre-trained model proposed by OpenAI, designed to learn a shared semantic embedding space for images and text. It is trained on large-scale natural image-text pairs, mapping images and text into the same-dimensional vector space using an image encoder and a text encoder. Contrastive learning is employed to pull corresponding image-text pairs closer in the embedding space while pushing non-corresponding pairs farther apart.

**ALBEF** [7] is a vision–language pre-trained model proposed by Salesforce, designed to enhance cross-modal semantic representations through an "align before fuse" strategy. Its core idea is to first align image and text features in a shared space via image–text contrastive learning, and then fuse them using a multimodal encoder to capture richer cross-modal interactions. Additionally, AL-BEF incorporates a momentum distillation mechanism, where a continuously updated momentum model generates pseudo-labels to improve training robustness. The model demonstrates strong performance on tasks such as image–text retrieval, visual question answering, and natural language visual reasoning, making it a key approach in the vision–language pretraining field.

---

[4] https://github.com/MILVLG/bottom-up-attention.pytorch
[5] https://nlp.stanford.edu/projects/glove/
[6] https://github.com/openai/CLIP
[7] https://github.com/salesforce/ALBEF

### I.4 MODEL DETAILS

In this section, we present the architectures of the models involved in the proposed MASK framework, as shown in Table 8. The $[B, M, 2048]$ indicates that a batch contains $B$ images, each image is divided into $M$ regions, and each region has a feature dimension of 2048. In our experiments, we set $B = 128$ and $M = 36$. It is worth noting that the model architectures are not fixed. In later sections, we will discuss how model size impacts the accuracy of image-text matching.

Table 8: Overview of the model architectures integrated into the MASK framework.

| Model | Layer | Input | Output | Params |
|---|---|---|---|---|
| $PAE$ | FC | $[B, M, 2048]$ | $[B, M, 512]$ | 1.05M |
| | ReLU | $[B, M, 512]$ | $[B, M, 512]$ | 0 |
| | Self-Attention | $[B, M, 512]$ | $[B, M, 512]$ | 1.05M |
| | Self-Attention | $[B, M, 512]$ | $[B, M, 512]$ | 1.05M |
| | Self-Attention | $[B, M, 512]$ | $[B, M, 512]$ | 1.05M |
| | Normalization | $[B, M, 512]$ | $[B, M, 512]$ | 1024 |
| $FRM$ | FC | $[B, M, 512]$ | $[B, M, 512]$ | 0.26M |
| | ReLU | $[B, M, 512]$ | $[B, M, 512]$ | 0 |
| | Self-Attention | $[B, M, 512]$ | $[B, M, 512]$ | 1.05M |
| | FC | $[B, M, 512]$ | $[B, M, 2048]$ | 1.05M |
| $MTM$ | FC | $[B, M, 512]$ | $[B, M, 300]$ | 0.15M |
| | ReLU | $[B, M, 300]$ | $[B, M, 300]$ | 0 |
| | Self-Attention | $[B, M, 300]$ | $[B, M, 300]$ | 0.37M |
| | Self-Attention | $[B, M, 300]$ | $[B, M, 300]$ | 0.37M |
| | FC | $[B, M, 300]$ | $[B, M, 300]$ | 0.09M |
| | ReLU | $[B, M, 300]$ | $[B, M, 300]$ | 0 |
| | Self-Attention | $[B, M, 300]$ | $[B, M, 300]$ | 0.37M |
| | FC | $[B, M, 300]$ | $[B, M, 300]$ | 0.09M |
| Total | | | | 8.1M |

## J ADDITIONAL EXPERIMENTS

### J.1 CROSS-DATASET IMAGE-TEXT MATCHING

Our proposed MASK can also enhance the generalization of conventional image-text matching models when applied to unseen datasets. Specifically, we try to re-rank conventional image-text matching models for the task of cross-dataset image-text matching. The experimental setup is as follows: (1) two representative image-text matching models (i.e., VSRN (Radford et al., 2021) and SAEM (Wu et al., 2019)) are trained on a source dataset (e.g., Flickr30K or MSCOCO), (2) these models are then evaluated on a different target dataset (e.g., MSCOCO or Flickr30K), and (3) using the proposed MASK to re-rank these models on the target dataset. It is important to note that neither the re-ranking methods nor the base models are trained on the target dataset.

Table 9: Cross-dataset image-text matching by re-ranking existing models on the Flickr30K and MSCOCO Datasets.

| Method | MSCOCO → Flickr30K | | | | | | | Flickr30K → MSCOCO | | | | | | |
|---|---|---|---|---|---|---|---|---|---|---|---|---|---|---|
| | Image Retrieval | | | Image Annotation | | | Rs | Image Retrieval | | | Image Annotation | | | Rs |
| | R@1 | R@5 | R@10 | R@1 | R@5 | R@10 | | R@1 | R@5 | R@10 | R@1 | R@5 | R@10 | |
| $VSRN$ | 42.3 | 69.3 | 78.1 | 53.1 | 79.5 | 87.5 | 409.9 | 14.0 | 31.7 | 42.2 | 20.4 | 40.0 | 50.0 | 198.4 |
| $VSRN + MACK_{2022}$ | 42.6 | 69.8 | 78.5 | 53.3 | 79.7 | 87.7 | 411.7 | 14.4 | 32.6 | 43.1 | 20.5 | 40.5 | 50.2 | 201.4 |
| $VSRN + LeaPRR_{2023}$ | 42.9 | 69.6 | 79.8 | 55.2 | 79.3 | 88.1 | 414.9 | 16.2 | 31.9 | 43.8 | 20.9 | 40.2 | 51.4 | 204.4 |
| $VSRN + MACK^{VG-M}_{2024}$ | 44.8 | 71.3 | 79.5 | 56.7 | 81.7 | 89.0 | 422.9 | 16.6 | 35.8 | 45.0 | 23.3 | 43.3 | 52.0 | 215.9 |
| $VSRN + FR_{2025}$ | 46.4 | 73.4 | 80.1 | 56.8 | 83.3 | 89.1 | 429.1 | 16.7 | 37.1 | 44.4 | 24.2 | 44.5 | 56.8 | 223.7 |
| $VSRN + \mathbf{MASK}$ | **47.3** | **73.7** | **83.9** | **57.9** | **83.6** | **89.9** | **436.3** | **17.6** | **38.7** | **45.9** | **26.7** | **44.8** | **58.7** | **232.4** |
| $SAEM$ | 41.4 | 70.2 | 80.0 | 53.4 | 80.9 | 89.6 | 415.5 | 14.8 | 34.0 | 45.0 | 23.2 | 45.4 | 57.4 | 219.8 |
| $SAEM + MACK_{2022}$ | 41.8 | 70.7 | 80.0 | 54.2 | 81.2 | 89.9 | 417.9 | 15.4 | 34.9 | 45.9 | 23.6 | 46.0 | 57.7 | 223.4 |
| $SAEM + LeaPRR_{2023}$ | 41.4 | 70.9 | 81.6 | 54.3 | 81.2 | 90.5 | 419.7 | 15.7 | 34.1 | 46.9 | 23.8 | 46.4 | 58.8 | 225.7 |
| $SAEM + MACK^{VG-M}_{2024}$ | 43.4 | 71.7 | 80.7 | 58.8 | 82.1 | 90.3 | 427.0 | 17.4 | 37.8 | 47.4 | 26.6 | 49.1 | 58.7 | 237.0 |
| $SAEM + FR_{2025}$ | 43.7 | 72.4 | 81.8 | 59.6 | 84.2 | 90.6 | 432.3 | 17.9 | 39.8 | 47.1 | 26.8 | 51.3 | 60.2 | 243.1 |
| $SAEM + \mathbf{MASK}$ | **45.1** | **72.6** | **82.0** | **61.6** | **86.9** | **91.9** | **440.1** | **19.1** | **40.5** | **47.8** | **27.6** | **53.9** | **62.3** | **251.2** |

In Table 9, the results of two kinds of cross-dataset image-text matching are both presented, which are explained as follows. MSCOCO $\rightarrow$ Flickr30K: training existing models on the MSCOCO dataset and testing them on the Flickr30K dataset. Flickr30K $\rightarrow$ MSCOCO: training existing models on the Flickr30K dataset and testing them on the MSCOCO dataset. We observe that applying MASK to re-rank the outputs of VSRN and SAEM consistently enhances their generalization performance on unseen datasets, with substantial relative improvements observed across both sub-tasks. For example, $VSRN + MASK$ performs much better than $VSRN$ by 5.0% and 4.8% in R@1 on the MSCOCO $\rightarrow$ Flickr30K task, and by 3.6% and 6.3% in R@1 on the Flickr30K $\rightarrow$ MSCOCO task. Compared with $VSRN + FR$, the relative improvements are also large, i.e., 7.2% $\sim$ 8.7% and 7.8% $\sim$ 8.1% in Rs when re-ranking VSRN and SAEM, respectively. Comparing the results in Table 9 with those in Table 2, we observe that the relative performance gains are more substantial for VSRN and SAEM, primarily because CLIP and ALBEF exhibit much higher baseline performance. In other words, re-ranking yields larger improvements when applied to less accurate models.

## J.2 VISUALIZATION OF POSITIVE AND NEGATIVE EXAMPLES FOR IMAGE RETRIEVAL AND IMAGE ANNOTATION

To better understand the OOD words, we show some representative examples of retrieved images or texts based on text or image queries by CLIP in Table 10. These examples are selected according to the following criteria: 1) **Positive examples** are those in which the ground-truth matched image or text is not ranked at top-1 by CLIP but is successfully promoted to a higher rank by MASK through re-ranking, and 2) **Negative examples** refer to situations where the ground-truth image or text is initially ranked at top-1 by CLIP but is pushed to a lower position after re-ranking by MASK. In the positive examples, it seems that the CLIP cannot well understand the semantic concepts such as "glasses", "pierced", and "broken". For instance, in the top-1 retrieved image for the first text query, there are no clear clues indicating "pierced" or "glasses", yet its rank is still higher than that of the ground-truth one. Similarly, in the top-1 retrieved text for the second image query, the annotation contains the word "glasses", even though the image itself does not include such information. While our MASK especially focuses on understanding these semantic concepts and can thus increase the corresponding similarity scores between the matched regions and words. However, MASK may also make incorrect decisions. For example, in the images retrieved for the fourth text query, MASK reduces the rank of the ground-truth image. This behavior can be attributed to the presence of adverbs (e.g., "very", "quite"), adjectives (e.g., "large", "excited"), and pronouns (e.g., "they", "this") in the text. Our further analysis indicates that the OOD words negatively affecting MASK are typically those that cannot correspond to specific visual regions. Attempting to construct region prototypes for such words introduces substantial semantic noise. Finally, it is important to note that negative examples contain not only non-visual adjectives, adverbs, and pronouns, but also some informative OOD words with tense or plural variations. Consequently, the final image-text matching accuracy is influenced by the combined effects of all these words.

## J.3 DIFFERENT DETECTOR COMPARISON

This work derives region representations using the Bottom-Up and Top-Down (BUTD) model, i.e., Faster-RCNN, which consists of a Region Proposal Network (RPN) (Ren et al., 2015) and a 101-layer Residual Network (ResNet101) (He et al., 2016) pretrained on the VG dataset. Since our constructed multimodal knowledge is also based on the VG dataset, this pretraining step is crucial for learning discriminative region representations and achieving strong performance. To validate this, we experiment with three alternative detectors within the proposed MASK framework. The first is DETR (Carion et al., 2020), a recently popular Transformer-based detector. The second is DINO (Zhang et al., 2023), evaluated in two versions: the original model and a variant pretrained on the VG dataset in the same manner as BUTD. The third is an enhanced version of BUTD, referred to as BUTD+, which employs ResNet152, ConvNeXt (Liu et al., 2022), and Swin Transformer (Liu et al., 2021) as the backbone networks to replace ResNet101.

We evaluate these detectors on the task of unpaired image-text matching and compare their performance on the Flickr30K dataset in Table 11. The results show that directly using either DETR or DINO leads to poor performance. This is primarily because they fail to extract region representations as accurately as BUTD when constructing multimodal knowledge. Specifically, BUTD uses Faster R-CNN as its backbone, a two-stage object detector that allows ground-truth bounding boxes

Table 10: Examples of retrieved top-3 images/texts based on text/image queries by CLIP. Groundtruth matched images are marked by black bounding boxes, which can be re-ranked higher by the proposed MASK. Similarly, the groundtruth text is annotated using the symbol "**(GT)**". The underlined words indicate the representative OOD words relative to the multimodal knowledge.

Table 11: Unpaired image-text matching using different detectors on the Flickr30K dataset.

| Detector | Architecture | Pretrained | Boxes | Image Retrieval | | | Image Annotation | | | Rs |
|---|---|---|---|---|---|---|---|---|---|---|
| | | | | R@1 | R@5 | R@10 | R@1 | R@5 | R@10 | |
| DETR | ResNet50+Transformer | No | 36 | 0.5 | 2.1 | 3.6 | 1.5 | 4.8 | 6.9 | 19.4 |
| DINO | ResNet50+Transformer | No | 36 | 0.8 | 2.7 | 4.2 | 1.9 | 4.6 | 7.4 | 21.6 |
| DINO | ResNet50+Transformer | Yes | 36 | 3.1 | 8.3 | 12.5 | 6.4 | 19.2 | 28.9 | 78.4 |
| BUTD | RPN + ResNet101 | Yes | 36 | 4.8 | 14.8 | 22.0 | 12.1 | 30.1 | 39.0 | 122.8 |
| BUTD+ | RPN + ResNet152 | Yes | 36 | 5.3 | 16.7 | 24.2 | 11.9 | 34.6 | 43.4 | 136.1 |
| BUTD+ | RPN + ResNet152 | Yes | 100 | 5.2 | 15.2 | 24.4 | 16.6 | 39.0 | 47.3 | 147.7 |
| BUTD+ | RPN + ConvNeXt | Yes | 36 | 5.7 | 18.0 | 26.1 | 12.9 | 37.4 | 46.9 | 147.0 |
| BUTD+ | RPN + Swin | Yes | 36 | 5.9 | 18.7 | 27.1 | 13.3 | 38.7 | 48.6 | 152.3 |

to be directly provided, yielding highly accurate feature representations. In contrast, DETR and DINO require generating hundreds of candidate bounding boxes and then selecting the one with the highest Intersection over Union (IoU) for each ground-truth box. This additional step inevitably introduces noise, thereby degrading the quality of the constructed knowledge and resulting in lower performance. In addition to inaccuracies in bounding box generation, the superior performance of

BUTD may also be attributed to its more powerful feature extraction network. To investigate this, we replace the ResNet101 backbone with ResNet152, ConvNeXt, and Swin Transformer, resulting in an enhanced version called BUTD+. This modification leads to a substantial performance improvement, demonstrating the benefit of stronger feature extraction.

## J.4 MODEL ARCHITECTURE ANALYSIS

This work acquires multimodal aligned knowledge based on three models: $PAE$, $FRM$, and $MTM$. Since the architecture of these models can significantly influence image-text matching accuracy, we use different architectures to perform the experiment of unpaired image-text matching and compare their performance on the Flickr30K dataset in Table 12.

We can see that increasing the number of fully connected (FC) layers leads to a decline in image-text matching performance, whereas adding more self-attention layers results in performance gains. For example, in the PAE model, increasing the number of fully connected layers from 1 to 3 results in a performance drop of approximately 19.4% in Rs on the Flickr30K dataset. When the number of layers is further increased from 3 to 5, accuracy decreases by an additional 11.3%. These findings demonstrate that FC layers have a substantial impact on model performance, and excessive depth can severely degrade accuracy. This can be explained as that FC layers apply fixed nonlinear transformations, where even minor training errors accumulate as the network deepens, gradually distorting the spatial distribution of features. In contrast, self-attention layers explicitly model similarity relationships among entities, inherently preserving or even reinforcing the semantic structure of the representations.

Table 12: Unpaired image-text matching using different model architectures on the Flickr30K dataset.

| Model | Layer | Params | Image Retrieval | | | Image Annotation | | | Rs |
|---|---|---|---|---|---|---|---|---|---|
| | | | R@1 | R@5 | R@10 | R@1 | R@5 | R@10 | |
| $PAE$ | 1 FC + 1 Self-Attention layer | 2.11M | 4.2 | 13.4 | 19.8 | 11.2 | 26.6 | 35.4 | 110.6 |
| | 1 FC + 3 Self-Attention layers | 4.21M | 4.8 | 14.8 | 22.0 | 12.1 | 30.1 | 39.0 | 122.8 |
| | 1 FC + 5 Self-Attention layers | 6.31M | 4.6 | 15.3 | 26.3 | 12.3 | 28.5 | 41.7 | 128.7 |
| | 3 FC + 3 Self-Attention layers | 4.73M | 2.6 | 11.8 | 16.4 | 10.3 | 26.1 | 36.2 | 103.4 |
| | 5 FC + 3 Self-Attention layers | 5.25M | 1.9 | 9.3 | 12.6 | 9.6 | 24.8 | 33.9 | 92.1 |
| $FRM$ | 1 FC + 1 Self-Attention layer | 2.1M | 4.2 | 13.9 | 21.2 | 11.7 | 29.6 | 37.5 | 118.1 |
| | 2 FC + 1 Self-Attention layer | 2.36M | 4.8 | 14.8 | 22.0 | 12.1 | 30.1 | 39.0 | 122.8 |
| | 3 FC + 1 Self-Attention layer | 2.62M | 3.7 | 12.6 | 20.2 | 10.8 | 28.7 | 36.1 | 112.1 |
| | 2 FC + 3 Self-Attention layers | 4.46M | 4.9 | 12.8 | 24.5 | 13.3 | 32.6 | 38.2 | 126.3 |
| | 2 FC + 5 Self-Attention layers | 6.56M | 4.6 | 15.3 | 24.2 | 13.9 | 33.8 | 38.9 | 130.7 |
| $MTM$ | 1 FC + 3 Self-Attention layers | 1.26M | 3.9 | 12.4 | 18.8 | 10.0 | 26.1 | 34.5 | 105.7 |
| | 3 FC + 3 Self-Attention layers | 1.44M | 4.8 | 14.8 | 22.0 | 12.1 | 30.1 | 39.0 | 122.8 |
| | 5 FC + 3 Self-Attention layers | 1.62M | 2.7 | 11.3 | 16.1 | 9.2 | 23.5 | 32.4 | 95.2 |
| | 3 FC + 1 Self-Attention layer | 0.7M | 4.2 | 13.6 | 20.4 | 11.7 | 28.2 | 37.5 | 115.6 |
| | 3 FC + 5 Self-Attention layers | 2.18M | 5.1 | 13.9 | 22.7 | 12.8 | 32.4 | 41.3 | 128.2 |

## J.5 DIFFERENT PRETRAINED WORD VECTORS COMPARISON

The semantic geometry of word embeddings directly influences the construction of OOD prototypes. To evaluate the impact of different pretrained word vectors on matching accuracy, we perform the experiment of unpaired image-text matching by MASK using different word vectors on the Flickr30K and MSCOCO datasets in Table 13. Experimental results show that GloVe consistently outperforms Word2Vec and FastText across both datasets, demonstrating its superior suitability for constructing OOD prototypes. Specifically, GloVe achieves 122.8% R@s on Flickr30K and 209.5% R@s on MSCOCO, surpassing Word2Vec by 2.7 $\sim$ 5.2% and FastText by 8.4 $\sim$ 14.4%. These performance differences can be attributed to the distinct characteristics of the word vectors. GloVe encodes global word–word co-occurrence statistics, enabling it to capture broader contextual relatedness. Such global semantic structure is crucial in cross-modal matching, where visual regions and textual words need to align through high-level associative semantics rather than strict synonymy. In contrast, Word2Vec, which learns from local context windows, excels at modeling fine-grained synonymy but is less capable of capturing the broader semantic relations required for cross-modal

alignment. FastText places greater emphasis on morphological similarity. However, morphological similarity does not necessarily imply semantic similarity, which introduces significant noise and ultimately reduces matching accuracy.

Table 13: Unpaired image-text matching by MASK using different pretrained word vectors on the Flickr30K and MSCOCO datasets.

| Word vectors | Flickr30K dataset | | | | | | | MSCOCO dataset | | | | | | |
| | Image Retrieval | | | Image Annotation | | | Rs | Image Retrieval | | | Image Annotation | | | Rs |
| | R@1 | R@5 | R@10 | R@1 | R@5 | R@10 | | R@1 | R@5 | R@10 | R@1 | R@5 | R@10 | |
|---|---|---|---|---|---|---|---|---|---|---|---|---|---|---|
| GloVe | **4.8** | **14.8** | **22.0** | **12.1** | **30.1** | **39.0** | **122.8** | **7.6** | **26.7** | **41.8** | **22.7** | **48.5** | **62.2** | **209.5** |
| Word2Vec | 4.6 | 14.0 | 21.3 | 11.3 | 28.6 | 37.8 | 117.6 | 7.4 | 26.2 | 41.5 | 22.2 | 47.8 | 61.7 | 206.8 |
| FastText | 4.2 | 12.6 | 19.9 | 10.0 | 26.1 | 35.6 | 108.4 | 7.0 | 25.4 | 40.7 | 21.3 | 46.3 | 60.4 | 201.1 |

### J.6 MODEL EFFICIENCY AND SIZE COMPARISON

As last, we compare the testing time and model size of our proposed MASK with those of CLIP, AL-BEF, and MACK on the same machine (Intel(R) Xeon(R) Platinum 8468V, 512 GB RAM memory, and 1 NVIDIA L40), based on their publicly available codes in Table 14. To eliminate the influence of other factors, such as the matching framework and feature extraction, we fix the batch size to 1 for all models during testing. In the table, the testing time is measured by seconds (s), and the model size is measured by the millions of model parameters (M).

From the table, we observe that MACK and MASK have significantly smaller model sizes compared to CLIP and ALBEF. The parameters of MACK come from the BUTD module used for extracting region representations. While the knowledge-based image-text matching is lightweight and requires no additional parameters. Similarly, for MASK, the majority of the testing time is attributed to the inference processes of the BUTD and $PAE$ models, which is considerably faster than the testing time required by CLIP and ALBEF.

Table 14: Model Efficiency and Size Comparison.

| Detector | Testing Times (s) | Model Size (M) |
|---|---|---|
| $CLIP$ | 0.08 | 291.0 |
| $ALBEF$ | 0.29 | 209.5 |
| $MACK$ | 0.03 | 42.5 |
| $MASK$ | 0.04 | 50.6 |

### J.7 HYPERPARAMETER ANALYSIS EXTENSION

To further evaluate the impact of different hyperparameters $\lambda_1$ and $\lambda_2$ on matching accuracy, we use different values of hyperparameters to perform the experiment of unpaired image-text matching and compare their performance on the Flickr30K and MSCOCO datasets in Table 15. The results indicate that MASK achieves the best performance in unpaired image-text matching when $\lambda_1 = \lambda_2 = 3$. This further indicates that the cross-modal alignment loss and the prototype consistency contrastive loss are complementary. By jointly optimizing the entire loss function in a balanced manner, the model's generalization capability can be substantially improved.

## K LIMITION AND FUTURE WORK

It is important to acknowledge certain limitations of the proposed MASK, which will be addressed in future work. First, the raw region representations are extracted using the pre-trained object detection model BUTD. It would be better to pretrain more advanced detectors on the VG dataset to provide more discriminative region presentations. Second, relying solely on nouns for unpaired image-text matching is suboptimal. It would be better to take all the other words into consideration for more accurate image-text matching.

Table 15: Unpaired image-text matching by MASK using different $\lambda_1$ and $\lambda_2$ on the Flickr30K and MSCOCO datasets.

| Hyperparameter | | Flickr30K dataset | | | | | | | MSCOCO dataset | | | | | | |
| | | Image Retrieval | | | Image Annotation | | | Rs | Image Retrieval | | | Image Annotation | | | Rs |
| | | R@1 | R@5 | R@10 | R@1 | R@5 | R@10 | | R@1 | R@5 | R@10 | R@1 | R@5 | R@10 | |
| $\lambda_1$ | 1.0 | 4.2 | 13.0 | 19.2 | 11.6 | 27.9 | 36.8 | 112.7 | 6.8 | 22.9 | 35.5 | 20.9 | 43.2 | 55.8 | 185.1 |
| | 3.0 | **4.8** | **14.8** | **22.0** | **12.1** | **30.1** | **39.0** | **122.8** | **7.6** | **26.7** | **41.8** | **22.7** | **48.5** | **62.2** | **209.5** |
| | 5.0 | 3.1 | 10.2 | 16.6 | 11.2 | 25.5 | 34.3 | 100.9 | 5.2 | 17.8 | 28.3 | 19.8 | 37.1 | 48.3 | 156.5 |
| $\lambda_2$ | 1.0 | 3.9 | 12.2 | 18.0 | 11.2 | 27.5 | 36.4 | 109.2 | 6.3 | 21.0 | 32.3 | 19.8 | 41.5 | 53.9 | 174.8 |
| | 3.0 | **4.8** | **14.8** | **22.0** | **12.1** | **30.1** | **39.0** | **122.8** | **7.6** | **26.7** | **41.8** | **22.7** | **48.5** | **62.2** | **209.5** |
| | 5.0 | 3.6 | 11.7 | 17.8 | 11.6 | 25.2 | 33.5 | 103.4 | 5.6 | 19.4 | 30.7 | 19.6 | 39.9 | 50.2 | 165.4 |

## L  THE USE OF LARGE LANGUAGE MODELS

We use LLMs solely to assist in checking grammatical correctness. After the initial check by the model, we further refine and correct any remaining grammatical issues manually. Therefore, the role of LLMs in this work is limited.

