# OpenReview forum: "Multimodal Aligned Semantic Knowledge for Unpaired Image-text Matching"
_ICLR.cc/2026/Conference — ICLR 2026 Oral_

### Official Review · Reviewer_EZrZ · 2025-10-26

**Soundness:** 3
**Presentation:** 2
**Contribution:** 2
**Rating:** 6
**Confidence:** 3

**Summary:**

This paper addresses an underexplored issue in cross-modal matching tasks: image-text matching without pairing. The authors propose a new method called MASK, whose core is to construct a cross-modal aligned semantic knowledge base. The model uses word embeddings as a bridge to match words with their prototypes, and by introducing prototype consistency contrastive learning loss to regulate the feature space, it overcomes the shortcomings of existing methods in handling out-of-distribution words and variance of visual representation distribution. Experiments on the Flickr30K and MSCOCO datasets show that MASK achieves leading performance in the unmatched matching task and can effectively serve as a re-ranking module to enhance the performance of existing pre-trained multimodal models.

**Strengths:**

1.	The framework proposed in this paper is intuitive and effective. It utilizes the semantic relationships of word embeddings to construct prototypes for out-of-distribution (OOD) vocabulary and a prototype consistency contrastive learning loss, clearly pointing out the key weaknesses of existing methods and having a clear motivation.
2.	This paper introduces external pre-trained word vectors as an auxiliary supervisory signal to establish an equivariant mapping that preserves the relationship between regional representations and word embeddings. This enables regional representations to effectively capture the semantic relationships between words.
3.	The experimental results on Flickr30K and MSCOCO show that MASK outperforms existing models and knowledge-based methods significantly in the image-text matching task, without paired images, proving its effectiveness.
4.	MASK can acquire a kind of logical semantic knowledge based on conceptual prototypes through structured learning. Uses MASK as an reordering enhancement plugin for CLIP and ALBEF models, demonstrating its effectiveness in prototype consistency contrastive learning and semantic relationship alignment.
5.	The appendix provides mathematical proofs for concepts such as peer-to-peer transformation mapping and the rationality of loss functions, enhancing the rigor and depth of the method. This is particularly valuable in applied research papers.

**Weaknesses:**

1.	The explanation of the OOD vocabulary processing mechanism is unclear. This is one of the core innovations of this paper. However, the description in Section 3.4 is overly mathematical and lacks an intuitive, step-by-step concrete example. For instance, when encountering a word that does not exist in the knowledge base, how can we use the Glove word vectors to find similar known words and weight them to synthesize their visual prototypes? The current explanation remains at the formula level, which is less readable and reduces the comprehensibility of the method.
2.	To construct the prototype of the OOD vocabulary, we need to sample m pairs of representations from the knowledge base. The paper does not explain at all how this sampling is carried out. Is it random sampling? Or is it Top-K sampling based on the semantic similarity with the OOD words? Different sampling strategies will greatly affect the quality of the final prototype, which is an important unspecified hyperparameter and must be clarified.
3.	The paper did not analyze the overlap between the vocabulary in the test set and the VG's 12,385 concepts. If the vast majority of nouns in the test set are already present in the knowledge base, then the challenge and performance improvement attributed to OOD words might be overestimated. It is necessary to clearly define the specific definition and proportion of OOD in the experiments.

**Questions:**

1.	Does the MASK method have good generalization ability on other types of datasets? For instance, how does the model perform on some datasets that contain more domain-specific words or more complex visual scenarios? Is there a plan to verify it on more diverse datasets?
2.	If certain words are not included in the pre-trained word vectors, or if the semantics of the words have changed in a specific domain, how will the MASK method handle this situation? Has there been any consideration of reducing reliance on the pre-trained word vectors or introducing other methods to obtain the semantic information of the words?

---

> ### Author Response · Authors · 2025-11-22
> **Authors' Rebuttal 1**
>
> Thanks a lot for your valuable time and constructive comments.
>
> **W1: The processing of OOD words is overly mathematical.**
>
> R1: Thank you for your constructive feedback. To enhance readability and ensure a clearer explanation of the OOD word processing pipeline, we have revised the presentation in Section 3.4 and streamlined the notation. We also incorporate a schematic illustration (Appendix B) to support intuitive understanding.
>
> When encountering OOD words that does not exist in the knowledge, we first obtain the word embedding of OOD words using GloVe. We then compute its similarity with all known word embeddings in the multimodal knowledge. Based on these similarities, we identify the top-$m$ most similar words. Finally, we construct the visual prototype vector for the OOD word by aggregating the visual prototype vectors of these top-$m$ words using their similarity scores as weights.
>
> ( Page 5, Line 248, Section 3.4; Page 14, Line 728, Appendix B. )
>
>
> **W2: How to sample $m$ paired multimodal representations?**
>
> R2: Thank you for the thoughtful comment! Given that the choice of $m$ has a significant impact on the semantic quality of the reconstructed prototypes, we provide additional clarification on the selection of $m$ in the last paragraph of Section 3.4. Specifically, we first obtain the word embedding of OOD words using GloVe. We then compute its similarity with all known word embeddings in the multimodal knowledge. Based on these similarities (sorted in descending order), we identify the top-$m$ most similar words. More details can be found in Section 3.4.
>
> ( Page 6, Line 277, Section 3.4. )
>
> In addition, we have included a sensitivity analysis of the $m$ in Table 5.
>
> |  $m$   | R1 (T2I)  | R5 (T2I)  | R10 (T2I) | R1 (I2T) | R5 (I2T)| R10  (I2T)| Rs (F30k) |  R1 (T2I)  | R5 (T2I)  | R10 (T2I) | R1 (I2T) | R5 (I2T)| R10  (I2T)| Rs (COCO) |
> |  ----  | ----  |  ----  | ----  |  ----  | ----  |  ----  | ----  |  ----  | ----  |  ----  | ----  |  ----  | ----  |  ----  |
> | 1     | 4.6 | 13.9  | 20.9 | 11.9  | 28.9 | 37.7  | 117.9 | 7.3  | 25.1 | 39.1  | 22.3 | 45.6  | 57.9 | 197.3  |
> | 10   | 4.8 | 14.8  | 22.0 | 12.1  | 30.1 | 39.0  | 122.8 | 7.6  | 26.7 | 41.8  | 22.7 | 48.5  | 62.2 | 209.5  |
> | 50   | 5.0 | 15.8  | 23.2 | 12.3  | 31.4 | 40.4  | 128.1 | 7.9  | 28.5 | 44.7  | 23.1 | 51.6  | 66.8 | 222.6  |
> | 100  | 4.9 | 15.3  | 22.6 | 12.2  | 30.7 | 39.7  | 125.4 | 7.7  | 27.6 | 43.3  | 22.9 | 50.0  | 64.5 | 216.0  |
>
>
> Experimental results show that the matching accuracy follows a rise–then–fall trend as the sampling size $m$ increases, achieving its optimum around $m$=50. This phenomenon can be explained as follows. When the sampling size is too small, the constructed visual prototype relies excessively on only a few nearest neighbors. Although this preserves strong local semantic characteristics, it also makes the prototype highly sensitive to noise and outliers in the word embedding space. As the sampling size increases to a moderate level, more semantically relevant neighbors contribute their visual information, thereby enhancing robustness and discriminability. However, when the sampling size becomes too large, semantically weak or marginal neighbors begin to dominate. Their less relevant visual cues dilute the contributions of the core semantic neighbors, ultimately reducing matching accuracy.
> For more details, please refer to Section 4.7.
>
> ( Page 9, Line 482, Section 4.7. )

---

> > ### Author Response · Authors · 2025-11-22
> > **Authors' Rebuttal 2**
> >
> > **W3: Definition and proportion of OOD**
> >
> > R3: Thanks for your insightful suggestion. The OOD words discussed in this paper are defined relative to the multimodal knowledge, rather than absolute OOD words in the traditional NLP sense. In the revised manuscript, we rewrite Section 3.4 and explicitly highlight that OOD words are defined relative to the multimodal knowledge, thereby providing a clearer understanding of the notion of OOD words.
> >
> > ( Page 5, Line 260, “Therefore, for OOD words relative to the knowledge…”. )
> >
> >
> > We did not quantify how many OOD words appear in the text descriptions of the test set, and it seems difficult to perform ablation studies based on the proportion of OOD words. However, we did set up a control group comparing whether OOD words are processed, as shown in Table 3. We observe that image-text matching accuracy significantly improves in both image retrieval and image annotation tasks when OOD words are incorporated, and this improvement is consistently validated across different datasets.
> >
> > |  Method   | R1 (T2I)  | R5 (T2I)  | R10 (T2I) | R1 (I2T) | R5 (I2T)| R10  (I2T)| Rs (F30k) |  R1 (T2I)  | R5 (T2I)  | R10 (T2I) | R1 (I2T) | R5 (I2T)| R10  (I2T)| Rs (COCO) |
> > |  ----  | ----  |  ----  | ----  |  ----  | ----  |  ----  | ----  |  ----  | ----  |  ----  | ----  |  ----  | ----  |  ----  |
> > | $MASK$   | 4.8 | 14.8  | 22.0 | 12.1  | 30.1 | 39.0  | 122.8 | 7.6  | 26.7 | 41.8  | 22.7 | 48.5  | 62.2 | 209.5  |
> > | $MASK\_{w/o \\; OOD}$    | 4.5 | 13.6 | 20.5 | 11.8  | 28.5 | 37.3  | 116.2 | 7.2 | 24.5 | 38.2  | 22.2 | 44.6  | 56.4 | 193.1 |
> >
> > ( Page 9, Line 432, Section 4.5. )
> >
> >
> > Furthermore, our further investigation reveals that OOD words do not always contribute positively to matching accuracy. As shown in Table 10, we present several negative examples and observe that certain non-visual adjectives (e.g., “large”, “excited”), adverbs (e.g., “very”, “quite”), or pronouns (e.g., “they”, “this”) tend to reduce matching accuracy. It would be more meaningful to separately quantify the proportion of OOD words that are positively or negatively aligned with matching accuracy. However, determining which OOD words positively influence matching performance remains an open question under ongoing investigation.
> >
> > ( Page 24, Line 1280, Appendix K.2. )

---

> > > ### Author Response · Authors · 2025-11-22
> > > **Authors' Rebuttal 3**
> > >
> > > **Q1: Generalization ability on other types of datasets**
> > >
> > > R4: Thank you for the thoughtful comment! MASK is constructed based on the VG dataset, and its matching accuracy is validated on the Flickr30k and MSCOCO datasets, which already demonstrates strong generalization capability. For domain-specific datasets, the knowledge can be fine-tuned to better adapt to the specific domain, a strategy that has been validated in published works [1]. We provide the details of the fine-tuning method in Appendix I for reference. In the future, we may explore knowledge fine-tuning strategies further and validate the effectiveness of our method on additional datasets.
> > >
> > > ( Page 21, Line 1107, Appendix I. )
> > >
> > > [1].Huang Y, Wang Y, Zeng Y, et al. Unpaired image-text matching via multimodal aligned conceptual knowledge[J]. IEEE Transactions on Pattern Analysis and Machine Intelligence, 2024.
> > >
> > >
> > > **Fine-Tuned Domain Knowledge:**
> > >
> > > To better adapt the multimodal aligned semantic knowledge to specific datasets, we can further fine-tune the prototypical region representations to obtain domain-specific knowledge. This fine-tuning step is optional and depends on the availability of unpaired data in the target dataset. Notably, the multimodal aligned semantic knowledge alone can also be applied directly and achieves strong performance. Given that annotating paired image-text data is costly, while unpaired images and texts are typically more accessible, we adopt a bidirectional region-word cycle-consistent learning approach in an unpaired learning setting.
> > >
> > > In particular, given a batch of unpaired images and texts, we first obtain a set of raw region representations $R = \{r_n\}\_{n=1, \dots, I} (R \in \mathcal{R}^{I \times M})$ using the pre-trained Faster-RCNN above and a set of parsed words through tokenization operation implemented via NLTK. Then, we use the knowledge as a cross-modal bridge to represent all the words into the corresponding prototypical region representations $u = \{ v_j \}_{j=1, \dots, J} (u \in \mathcal{R}^{J \times Z} )$. For the set of regions $R$, we extract region representations $\mu \in \mathcal{R}^{I \times Z}$ by utilizing the $PAE$ model $h$.
> > >
> > > We utilize a bidirectional region-word cycle-consistent loss to learn a parametric transformation matrix $W \in \mathcal{R}^{Z \times Z}$ \footnote{We also stack multiple transformation matrices in a nonlinear way, but find it does not necessarily lead to better performance}. This loss incorporates two cross-modal similarity measurement processes: region-to-word (R2W) and word-to-region (W2R). In the R2W process, similarities between each word and all regions are first computed, and these similarities are then used as weights to aggregate all regions into a reconstructed word representation. Conversely, in the W2R process, similarities between each region and all words are computed to reconstruct regions from the word representations. The corresponding formulations are
> > >
> > > \begin{equation}
> > > 	S = uW (\mu W)^\top,
> > > \end{equation}
> > >
> > > \begin{equation}
> > > 	\hat{u} = softmax(S) \; \mu W, \; \hat{\mu} = softmax(S^\top) \; uW,
> > > \end{equation}
> > >
> > > where $S \in \mathcal{R}^{J \times I}$ is the similarity matrix between transformed word and region representations, $\hat{u} \in \mathcal{R}^{J \times Z}$ contains the reconstructed word representations from region representations, and $\hat{\mu} \in \mathcal{R}^{I \times Z}$ contains the reconstructed region representations from word representations.
> > >
> > >
> > >
> > > Each original word (or region) representation is then compared with its reconstructed counterpart to determine whether they correspond to the same entity. By minimizing the cross-entropy between the predicted and ground-truth labels, we obtain a self-supervised loss, $\mathcal{L}_{ss}$, which is used to optimize $W$:
> > >
> > > \begin{equation}
> > > 	\hat{Y}^{R2W} = softmax(uW \hat{u}^\top)^\top, \; \hat{Y}^{W2R} = softmax(\mu W \hat{\mu}^\top)^\top,
> > > \end{equation}
> > >
> > > \begin{equation}
> > > 	\mathcal{L}^{R2W} = -\sum_{j=1}^{J} y_j^\top log(\hat{y}_j),
> > > \end{equation}
> > >
> > > \begin{equation}
> > > 	\mathcal{L}^{W2R} = -\sum_{n=1}^{I} y_n^\top log(\hat{y}_n),
> > > \end{equation}
> > >
> > > \begin{equation}
> > > 	\mathcal{L}^{ss} = \mathcal{L}^{R2W} + \mathcal{L}^{W2R},
> > > \end{equation}
> > >
> > > where $\hat{Y}^{R2W} \in \mathcal{R}^{J \times J}$ and $\hat{Y}^{W2R} \in \mathcal{R}^{I \times I}$ are two matrices including the predicted labels in R2W and W2R directions, respectively. In $\hat{Y}^{R2W}$ and $\hat{Y}^{W2R}$, the $j$-th and $n$-th columns are denoted as $\hat{y}_j$ and $\hat{y}_n$, respectively. $y_j$ and $y_n$ are two groundtruth label vectors, whose the $j$-th and $n$-th values are ones and the rest are zeros. After the cycle consistent learning, we can use the learnable $W$ to transform all prototypical region representations into the fine-tuned domain knowledge, denoted as $\{ (w_k, \hat{v}_k  ) \}\_{k=1, \dots, K}$, where $\hat{v}\_k = v\_k W \in \mathcal{R}^Z$.

---

> > > > ### Author Response · Authors · 2025-11-22
> > > > **Authors' Rebuttal 4 (End)**
> > > >
> > > > **Q2: Certain words are not included in the pre-trained word vectors**
> > > >
> > > > R5: Thank you for your constructive feedback. GloVe (glove-840B-300d), for example, is trained on 840 billion tokens and builds a vocabulary of 2.2 million words, covering nearly all common words. Therefore, we do not need to be overly concerned about the potential performance degradation caused by words that are not covered in the pretrained word embeddings. If the semantics of certain words shift within a specific domain, we recommend fine-tuning the knowledge to better adapt to the domain-specific data ( Appendix I).
> > > >
> > > > ( Page 21, Line 1107, Appendix I. )
> > > >
> > > >
> > > > Not relying on any pre-trained word vectors implies that the model must learn the semantic representations of words from scratch. Masked Language Modeling (MLM) is a commonly adopted strategy for this purpose. Specifically, a portion of tokens (e.g., 15%) in large-scale unlabeled text corpora is randomly selected for masking: most masked tokens are replaced with a special symbol [MASK], while a small fraction is replaced by random words or left unchanged to prevent the model from simply learning a spurious “see-MASK-then-predict” pattern. The corrupted sentences are then fed into a Transformer encoder, which is trained to predict the original tokens based on their context. The objective minimizes the divergence between the predicted tokens and the ground-truth tokens. By repeating this process over massive corpora, the model progressively learns linguistic structure, semantic relations, and contextual dependencies, ultimately producing word representations. However, surpassing high-quality pre-trained embeddings is extremely challenging. Such a model must satisfy several demanding conditions:
> > > >
> > > > (1) access to sufficiently large and diverse training corpora;
> > > >
> > > > (2) a model with enough capacity;
> > > >
> > > > (3) a well-designed training objective; and
> > > >
> > > > (4) extensive training to full convergence.
> > > >
> > > > In practice, models trained entirely from scratch often underperform pre-trained word embeddings, which already capture rich semantic information learned from extremely large text corpora.

---

> > > > > ### Comment · Reviewer_EZrZ · 2025-11-25
> > > > >
> > > > > The authors have provided detailed explanations and sensitivity analyses of the Top-m sampling strategy, clarified the definition of OOD vocabulary and quantitatively verified its impact, as well as further discussed the generalization ability of the model. These efforts have effectively addressed most of the key issues I initially raised. Based on the contributions of this paper, I decide to maintain the rating scores.

---

### Official Review · Reviewer_VshV · 2025-10-29

**Soundness:** 3
**Presentation:** 3
**Contribution:** 3
**Rating:** 6
**Confidence:** 4

**Summary:**

The paper proposes MASK (Multimodal Aligned Semantic Knowledge), a framework for unpaired image-text matching that leverages external semantic knowledge to bridge the gap between textual and visual modalities.
Instead of relying on paired image-text data, the method constructs visual prototypes and uses pretrained word embeddings to generate prototypes for out-of-distribution (OOD) words through a weighted combination of existing ones.
A prototype-consistency contrastive loss is further introduced to reduce intra-class variance and enhance alignment stability between textual and visual representations.
Extensive experiments on Flickr30k and MSCOCO demonstrate competitive retrieval performance and validate the benefit of incorporating external knowledge.
The approach can also serve as a lightweight re-ranking module to enhance existing vision-language models.

**Strengths:**

1. This paper proposes a novel cross-modal semantic alignment method, MASK, which constructs representative prototypes for OOD words by exploiting the intrinsic relationships amongword embeddings.

2. The paper provides solid comparisons with both model-based and knowledge-based baselines on Flickr30k and MSCOCO, together with ablation studies, sensitivity analysis, and visualization of learned representations.
3. The proposed method can serve as a lightweight plug-in re-ranking module, enhancing existing vision-language models such as CLIP or ALBEF without retraining them end-to-end.

4. This paper introduce a prototype consistency contrastive learning loss to structurally regularize the feature space, mitigating the adverse impact of distributional variance.

**Weaknesses:**

1. Over-strong assumption and circular reasoning in the OOD prototype proof
The theoretical analysis in Appendix B relies on a strong linear/isometric assumption that equates textual and visual prototype spaces. Moreover, the proof assumes the existence of a relation-preserving mapping
f,which is exactly what the model is trained to learn—thus creating a circular argument. Additional experiments or a more rigorous justification are needed to support this claim.

2. High dependence on region proposals
The method heavily relies on the quality of the region proposals provided by upstream object detectors. The experiments show large performance gaps between different detectors (e.g., BUTD vs. DETR/DINO), indicating that the framework’s robustness and generalization depend strongly on detector performance.

3. Insufficient analysis of pretrained word embeddings
The approach uses pretrained word vectors to construct OOD prototypes, but the effect of different embeddings (e.g., GloVe, word2vec, fastText) is not explored.  Since the semantic geometry of word embeddings directly affects OOD prototype generation, additional ablation or sensitivity analysis would strengthen the paper.

4. Missing discussion on the parameter m for OOD prototype construction.
In Eq.(8), OOD prototypes are computed by weighting m known prototypes, but the paper does not explain how m is chosen or provide related experiments. The selection of m could significantly affect the semantic quality of constructed prototypes.
Including an analysis or sensitivity study on this parameter would help clarify its impact and strengthen the conclusion.

**Questions:**

See the weakness.

---

> ### Author Response · Authors · 2025-11-22
> **Authors' Rebuttal 1**
>
> We greatly appreciate your detailed comments. We hope our response below effectively addresses your concerns.
>
> **W1: The Circular Dependency Between the Isometry Assumption and the Training Loss of Model $f$.**
>
> R1: Thank you for your thoughtful review. According to Appendix B (now Appendix C), the isometry assumption is beneficial for constructing high-quality prototype vectors for OOD words. However, this isometry assumption is expected to be preserved by the $MTM$ model $f$. To examine whether training $f$ indeed facilitates the construction of reliable OOD prototypes, we decompose the problem into two parts:
>
> 1.Proof: The objective existence of the Isometric Hypothesis (Appendix E).
>
> 2.Proof: $\mathcal{L}\_{cm}$ encourages $f$ to approach the Isometry Assumption (Appendix F).
>
> ( Page 17, Line 890, Appendix E;  Page 18, Line 934, Appendix F.)
>
>
>
> **1.Proof: The objective existence of the Isometric Hypothesis.**
>
> Proof: For any finite set of region representations $\{  \mu_i\}_{i=1}^n$, there necessarily exists a mapping $f$ in an appropriate Euclidean space such that:
>
> \begin{equation}
> 	d_s(f(\mu_i; \Theta_f), f(\mu_j; \Theta_f)) \propto d_s(\mu_i, \mu_j).
> \end{equation}
>
> Given $n$ non-zero vectors $ \mu_1 , \dots, \mu_n \in \mathcal{R}^Z$, normalize them to obtain:
>
> \begin{equation}
> 	\hat{\mu}_i  := \frac{\mu_i}{\left \| \mu_i \right \|_2} , i=1, \dots, n .
> \end{equation}
>
> We aim to construct a set of vectors $\{ y_i \}_{i=1}^n$ in a Euclidean space as $f(\mu_i)$, and demonstrate that $cos(y_i, y_j) = cos(\mu_i, \mu_j)$ can be achieved.
>
> Next, we organize the pairwise similarities into a matrix, facilitating the use of spectral decomposition to construct vectors that satisfy the given inner product relationships. We define the $n \times n$ Gram matrix $G$ as follows:
>
> \begin{equation}
> 	G_{ij} := cos(\hat{\mu}_i, \hat{\mu}_j) = \hat{\mu}_i^\top \hat{\mu}_j.
> \end{equation}
>
> Here $G$ is a real symmetric positive semi-definite (PSD) matrix, so for any vector $z \in \mathcal{R}^n$, it satisfies:
>
> \begin{equation}
> 	z ^ \top  G z = \left \|  {\textstyle \sum_{i}^{} z_i \hat{\mu}_i}  \right \| _2^2 \ge 0,
> \end{equation}
>
> where the $G_{ii} = 1$. According to the properties of real symmetric PSD matrix, there exist an orthogonal matrix $\mathcal{U} \in \mathcal{R}^{n \times n}$ and a diagonal matrix $\Lambda = diag(\lambda_1, \dots \lambda_n) (\lambda_k \ge 0)$ such that:
>
> \begin{equation}
> 	G = \mathcal{U} \Lambda \mathcal{U}^ \top,
> \end{equation}
>
> where $r = rank (G) \le n$. Take the non-zero eigenvalue part of $ \Lambda $ as $ \Lambda_r \in \mathcal{R}^{r \times r}$, and take the first $r$ columns of the corresponding eigenvectors to obtain $\mathcal{U}_r \in \mathcal{R}^{n \times r}$:
>
> \begin{equation}
> 	G = \mathcal{U}_r \Lambda_r \mathcal{U}_r^ \top,
> \end{equation}
>
> \begin{equation}
> 	Y := \mathcal{U}_r \Lambda_r^{1/2} \in \mathcal{R}^{n \times r},
> \end{equation}
>
> where we write $Y$ in row-vector form, and denote its $i$-th row as $y_i ^\top$ ($y_i \in \mathcal{R}^r$):
>
> \begin{equation}
> 	(Y Y^\top)_{ij} = y_i ^\top y_j = { (\mathcal{U}_r \Lambda _r^{1/2})(\mathcal{U}_r \Lambda _r^{1/2}) ^ \top } _{ij} = {\mathcal{U}\_r \Lambda\_r \mathcal{U}\_r^ \top}\_{ij} = G\_{ij}.
> \end{equation}
>
>
> Therefore, we have constructed $n$ vectors $y_1, \dots, y_n \in \mathcal{R}^r$ that satisfy the inner-product relation $y_i ^\top y_j = G_{ij}$:
>
> \begin{equation}
> 	\left \| y_i \right \| ^2_2 = y_i ^ \top  y_i = G_{ii} = 1,
> \end{equation}
>
> \begin{equation}
> 	cos(y_i, y_j) = \frac{y_i ^ \top  y_j}{\left \| y_i \right \| \left \| y_j \right \|} = \frac{G_{ij}}{1 \cdot  1} = cos(\hat{\mu}_i, \hat{\mu}_j),
> \end{equation}
>
> \begin{equation}
> 	cos(y_i, y_j) = cos(\hat{\mu}_i, \hat{\mu}_j) = cos(\mu_i, \mu_j).
> \end{equation}
>
> By setting $f(\mu _i) := y_i  (i=1, \dots, n)$, we obtain:
>
> \begin{equation}
> 	cos(f(\mu_i), f(\mu_j)) = cos(\mu_i, \mu_j) \forall i,j,
> \end{equation}
>
> \begin{equation}
> 	d_s(f(\mu_i; \Theta_f), f(\mu_j; \Theta_f)) \propto d_s(\mu_i, \mu_j).
> \end{equation}

---

> > ### Author Response · Authors · 2025-11-22
> > **Authors' Rebuttal 2**
> >
> > **2. Proof: $\mathcal{L}_{cm}$ (Eq. (9)) Encourages $f$ to Approach the Isometry Assumption (Eq. (8)).**
> >
> > Given the pre-trained word embeddings $w_i$ and the predicted word embeddings $w_i' = f(\mu_i; \Theta_f)$, we normalize them to obtain:
> >
> > \begin{equation}
> > 	\hat{w}_i' := \frac{w_i'}{\left \| w_i' \right \|}, \hat{w}_i := \frac{w_i}{\left \| w_i \right \|}.
> > \end{equation}
> >
> > Similarly, we normalize the region representations to obtain $ \hat{\mu}_i := \frac{\mu_i}{\left \| \mu_i \right \|} $.
> >
> > $\mathcal{L}\_{cm}$ consists of $\mathcal{L}\_{word}$ and $\mathcal{L}\_{struct}$ (Appendix G). For a given batch size $B$, there exist constants $\epsilon_w > 0$ and $\epsilon_s > 0$ such that,
> >
> > when $\mathcal{L}\_{word} \le \epsilon_w$ and $\mathcal{L}_{struct} \le \epsilon_s$, the following inequality holds:
> >
> > \begin{equation}
> > 	\left | cos(\hat{w}_i',  \hat{w}_j') - cos(\mu_i, \mu_j) \right |  \le 4 \sqrt{2 \epsilon_w } + \sqrt{\frac{ \epsilon_s
> > 		}{M} },
> > \end{equation}
> >
> > where $M$ denotes the number of ordered pairs ($i \ne j$), and $i, j \in B$.
> >
> > When $\epsilon_w \to 0$ and $\epsilon_s \to 0$, $cos(\hat{w}_i',  \hat{w}_j') - cos(\mu_i, \mu_j)$ admits an upper bound arbitrarily close to 0,
> >
> > meaning that the cosine similarity between the mapped vector pairs approaches that in the original $\mu$-space,
> >
> > thereby achieving $d_s(f(\mu_i; \Theta_f), f(\mu_j; \Theta_f)) \propto d_s(\mu_i, \mu_j)$.
> >
> > We first convert the point-wise cosine error into the Euclidean difference between vectors. According to Appendix D, we have:
> >
> > \begin{equation}
> > 	\left \| \hat{w}_i' - \hat{w}_i \right \| _2 ^2 = 2 (1 - cos(\hat{w}_i',  \hat{w}_i)).
> > \end{equation}
> >
> > For each case where $i \in B$, let $\delta_i := 1 - cos(\hat{w}_i',  \hat{w}_i) \ge  0$. Then:
> >
> > \begin{equation}
> > 	\left \| \hat{w}_i' - \hat{w}_i \right \| _2 =\sqrt{2 \delta _i} ,
> > \end{equation}
> >
> > \begin{equation}
> > 	\sum_{i \in B}^{} \delta \_i = \mathcal{L}\_{word} \le \epsilon _w  ,
> > \end{equation}
> >
> > \begin{equation}
> > 	\delta _i \le  \epsilon _w  \Rightarrow \left \| \hat{w}_i' - \hat{w}_i \right \| _2 \le \sqrt{2 \epsilon _w}.
> > \end{equation}
> >
> >
> > To quantify how point-wise alignment errors influence the pairwise cosine discrepancies,
> >
> > we characterize the variation of individual entries in the cosine similarity matrix using the norm difference. Then:
> >
> > \begin{equation}
> > 	\left | cos(\hat{w}_i',  \hat{w}_j') - cos(\hat{w}_i, \hat{w}_j) \right | = \left | \hat{w}_i' \cdot \hat{w}_j'  - \hat{w}_i  \cdot  \hat{w}_j   \right |,
> > \end{equation}
> >
> > \begin{equation}
> > 	\begin{aligned}
> > 		& \left | \hat{w}_i' \cdot \hat{w}_j'  - \hat{w}_i  \cdot  \hat{w}_j   \right | \\ & = \left | (\hat{w}_i' - \hat{w}_i) \cdot  \hat{w}_j'  + \hat{w}_i \cdot  (\hat{w}_j' - \hat{w}_j) \right |  \\ &  \le \left \| \hat{w}_i' - \hat{w}_i \right \| _2 \cdot  \left \| \hat{w}_j' \right \|_2 + \left \| \hat{w}_i \right \|_2  \cdot   \left \|  \hat{w}_j' - \hat{w}_j \right \| _2  \\  & \le \left \| \hat{w}_i' - \hat{w}_i \right \| _2  +  \left \| \hat{w}_j' - \hat{w}_j \right \| _2
> > 	\end{aligned}
> > \end{equation}
> >
> > \begin{equation}
> > 	\left | \hat{w}_i' \cdot \hat{w}_j'  - \hat{w}_i  \cdot  \hat{w}_j   \right | \le \sqrt{2 \epsilon _w} + \sqrt{2 \epsilon _w} = 2 \sqrt{2 \epsilon _w}.
> > \end{equation}
> >
> > To quantify the influence of the structural loss $\mathcal{L}\_{struct}$ on the reconstruction error,
> >
> > we convert the structural loss into an upper bound on the per-pair error.
> >
> > Let the per-pair error be defined as $e_{ij} := cos(\hat{w}_i',  \hat{w}_j') - cos(\mu_i, \mu_j)$. Then we obtain:
> >
> > \begin{equation}
> > 	\sum_{i \ne j}^{} e_{ij}^2 = \mathcal{L}_{struct} \le \epsilon _s ,
> > \end{equation}
> >
> > \begin{equation}
> > 	\max_{i \ne j} \left | e_{ij} \right | \le \sqrt{\sum_{i \ne j}^{} e_{ij}^2} \le \sqrt{\epsilon _s} .
> > \end{equation}
> >
> > If we distribute $\epsilon _s$ uniformly across all pairs, then the absolute error of each pair is bounded above by $\epsilon _s / M$. Then:
> >
> > \begin{equation}
> > 	\begin{aligned}
> > 		& \left | cos(\hat{w}_i',  \hat{w}_j') - cos(\mu_i, \mu_j) \right | \\ & = \left | cos(\hat{w}_i',  \hat{w}_j') - cos(\hat{w}_i,  \hat{w}_j) + cos(\hat{w}_i,  \hat{w}_j) - cos(\mu_i, \mu_j) \right |  \\ &  \le \left |    cos(\hat{w}_i',  \hat{w}_j') - cos(\hat{w}_i,  \hat{w}_j) \right |  + \left |  cos(\hat{w}_i,  \hat{w}_j) - cos(\mu_i, \mu_j)    \right |   \\  & \le  2 \sqrt{2 \epsilon _w} + \left |    cos(\hat{w}_i,  \hat{w}_j) - cos(\hat{w}_i',  \hat{w}_j') + cos(\hat{w}_i',  \hat{w}_j') - cos(\mu_i, \mu_j) \right |  \\ & \le 4 \sqrt{2 \epsilon_w } + \sqrt{\frac{ \epsilon_s
> > 			}{M}}
> > 	\end{aligned}
> > \end{equation}
> >
> >
> > Therefore, by enforcing $\epsilon _w \to 0$ and $\epsilon _s \to 0$ during training, we obtain the following expression:
> >
> > \begin{equation}
> > 	\left | cos(\hat{w}_i',  \hat{w}_j') - cos(\mu_i, \mu_j) \right | \to 0 .
> > \end{equation}
> >
> > \begin{equation}
> > 	d_s(f(\mu_i; \Theta_f), \; f(\mu_j; \Theta_f)) \propto d_s(\mu_i, \mu_j) .
> > \end{equation}

---

> > > ### Author Response · Authors · 2025-11-22
> > > **Authors' Rebuttal 3**
> > >
> > > **W2: The Impact of Region Encoders on Matching Accuracy.**
> > >
> > > R2: Thank you for your constructive feedback. We agree with your point. The matching accuracy is affected by the choice of the region encoder. To further evaluate the impact of different region encoders on image-text matching accuracy, we additionally incorporated ConvNeXt [1] and Swin Transformer [2] into the ablation studies in Table 11. We can observe that replacing the backbone with a more powerful one indeed improves the final image-text matching accuracy. For more details, please refer to Appendix K.3.
> > >
> > > ( Page 25, Line 1342, Appendix K.3. )
> > >
> > > |  Detector   | Architecture | Pretrained  | Boxes  | R1 (T2I)  | R5 (T2I)  | R10 (T2I) | R1 (I2T) | R5 (I2T)| R10  (I2T)| Rs  |
> > > |  ----  | ----  | ----  | ----  | ----  | ----  | ----  | ----  | ----  | ----  | ----  |
> > > | DETR     | ResNet50+Transformer | No  | 36   | 0.5 | 2.1   | 3.6 | 1.5 | 4.8 | 6.9 | 19.4 |
> > > | DINO      | ResNet50+Transformer | No  | 36   | 0.8 | 2.7   | 4.2 | 1.9 | 4.6| 7.4 | 21.6 |
> > > | DINO      | ResNet50+Transformer | Yes | 36   | 3.1 | 8.3   | 12.5 | 6.4 | 19.2 | 28.9 | 78.4 |
> > > | BUTD     | RPN+ResNet101           | Yes | 36   | 4.8 | 14.8 | 22.0 | 12.1 | 30.1 | 39.0 | 122.8 |
> > > | BUTD+   | RPN+ResNet152           | Yes | 36   | 5.3 | 16.7 | 24.2 | 11.9 | 34.6 | 43.4 | 136.1 |
> > > | BUTD+   | RPN+ResNet152           | Yes | 100 | 5.2 | 15.2 | 24.4 | 16.6 | 39.0 | 47.3 | 147.7 |
> > > | BUTD+   | RPN+ConvNeXt             | Yes | 36   | 5.7 | 18.0 | 26.1 | 12.9 | 37.4 | 46.9 | 147.0 |
> > > | BUTD+   | RPN+Swin                     | Yes | 36   | 5.9 | 18.7  | 27.1 | 13.3 | 38.7 | 48.6 | 152.3 |
> > >
> > >
> > > [1]. Liu Z, Mao H, Wu C Y, et al. A convnet for the 2020s[C]//Proceedings of the IEEE/CVF conference on computer vision and pattern recognition. 2022: 11976-11986.
> > >
> > > [2]. Liu Z, Lin Y, Cao Y, et al. Swin transformer: Hierarchical vision transformer using shifted windows[C]//Proceedings of the IEEE/CVF international conference on computer vision. 2021: 10012-10022.
> > >
> > >
> > >
> > >
> > > **W3: Different Pretrained Word Vectors Comparison**
> > >
> > > Q3: Thank you for the thoughtful comment! The semantic geometry of word embeddings directly influences the construction of OOD prototypes. To evaluate the impact of different pretrained word vectors on matching accuracy, we perform the experiment of unpaired image-text matching by MASK using different word vectors on the Flickr30k and MSCOCO datasets in Table 13.
> > >
> > >
> > > |  Word vectors   | R1 (T2I)  | R5 (T2I)  | R10 (T2I) | R1 (I2T) | R5 (I2T)| R10  (I2T)| Rs (F30k) |  R1 (T2I)  | R5 (T2I)  | R10 (T2I) | R1 (I2T) | R5 (I2T)| R10  (I2T)| Rs (COCO) |
> > > |  ----  | ----  |  ----  | ----  |  ----  | ----  |  ----  | ----  |  ----  | ----  |  ----  | ----  |  ----  | ----  |  ----  |
> > > | GloVe     | 4.8 | 14.8  | 22.0 | 12.1  | 30.1 | 39.0  | 122.8 | 7.6  | 26.7 | 41.8  | 22.7 | 48.5  | 62.2 | 209.5  |
> > > | Word2Vec   | 4.6 | 14.0  | 21.3 | 11.3  | 28.6 | 37.8  | 117.6 | 7.4  | 26.2 | 41.5  | 22.2 | 47.8  | 61.7 | 206.8  |
> > > | FastText   | 4.2 | 12.6  | 19.9 | 10.0  | 26.1 | 35.6  | 108.4 | 7.0  | 25.4 | 40.7  | 21.3 | 46.3  | 60.4 | 201.1  |
> > >
> > >
> > >
> > > Experimental results show that GloVe consistently outperforms Word2Vec and FastText across both datasets, demonstrating its superior suitability for constructing OOD prototypes. Specifically, GloVe achieves 122.8\% R@s on Flickr30k and 209.5\% R@s on MSCOCO, surpassing Word2Vec by 2.7 $\sim$ 5.2\% and FastText by 8.4 $\sim$ 14.4\%. These performance differences can be attributed to the distinct characteristics of the word vectors. GloVe encodes global word–word co-occurrence statistics, enabling it to capture broader contextual relatedness. Such global semantic structure is crucial in cross-modal matching, where visual regions and textual words need to align through high-level associative semantics rather than strict synonymy. In contrast, Word2Vec, which learns from local context windows, excels at modeling fine-grained synonymy but is less capable of capturing the broader semantic relations required for cross-modal alignment. FastText places greater emphasis on morphological similarity. However, morphological similarity does not necessarily imply semantic similarity, which introduces significant noise and ultimately reduces matching accuracy. For more details, please refer to Appendix K.5.
> > >
> > > ( Page 26, Line 1397, Appendix K.5. )

---

> > > > ### Author Response · Authors · 2025-11-22
> > > > **Authors' Rebuttal 4 (End)**
> > > >
> > > > **W4: Missing discussion on the parameter m for OOD prototype construction.**
> > > >
> > > > R4: Thank you for the thoughtful comment! Given that the choice of $m$ has a significant impact on the semantic quality of the reconstructed prototypes, we provide additional clarification on the selection of $m$ in the last paragraph of Section 3.4. Specifically, we first obtain the word embedding of OOD words using GloVe. We then compute its similarity with all known word embeddings in the multimodal knowledge. Based on these similarities, we identify the top-m most similar words. Finally, we construct the visual prototype vector for the OOD word by aggregating the visual prototype vectors of these top-m words using their similarity scores as weights. More details can be found in Section 3.4.
> > > >
> > > > ( Page 6, Line 277, Section 3.4. )
> > > >
> > > > In addition, we have included a sensitivity analysis of the $m$ in Table 5.
> > > >
> > > > |  $m$   | R1 (T2I)  | R5 (T2I)  | R10 (T2I) | R1 (I2T) | R5 (I2T)| R10  (I2T)| Rs (F30k) |  R1 (T2I)  | R5 (T2I)  | R10 (T2I) | R1 (I2T) | R5 (I2T)| R10  (I2T)| Rs (COCO) |
> > > > |  ----  | ----  |  ----  | ----  |  ----  | ----  |  ----  | ----  |  ----  | ----  |  ----  | ----  |  ----  | ----  |  ----  |
> > > > | 1     | 4.6 | 13.9  | 20.9 | 11.9  | 28.9 | 37.7  | 117.9 | 7.3  | 25.1 | 39.1  | 22.3 | 45.6  | 57.9 | 197.3  |
> > > > | 10   | 4.8 | 14.8  | 22.0 | 12.1  | 30.1 | 39.0  | 122.8 | 7.6  | 26.7 | 41.8  | 22.7 | 48.5  | 62.2 | 209.5  |
> > > > | 50   | 5.0 | 15.8  | 23.2 | 12.3  | 31.4 | 40.4  | 128.1 | 7.9  | 28.5 | 44.7  | 23.1 | 51.6  | 66.8 | 222.6  |
> > > > | 100  | 4.9 | 15.3  | 22.6 | 12.2  | 30.7 | 39.7  | 125.4 | 7.7  | 27.6 | 43.3  | 22.9 | 50.0  | 64.5 | 216.0  |
> > > >
> > > >
> > > >
> > > > Experimental results show that the matching accuracy follows a rise-then-fall trend as the sampling size $m$ increases, achieving its optimum around $m$=50. This phenomenon can be explained as follows. When the sampling size is too small, the constructed visual prototype relies excessively on only a few nearest neighbors. Although this preserves strong local semantic characteristics, it also makes the prototype highly sensitive to noise and outliers in the word embedding space. As the sampling size increases to a moderate level, more semantically relevant neighbors contribute their visual information, thereby enhancing robustness and discriminability. However, when the sampling size becomes too large, semantically weak or marginal neighbors begin to dominate. Their less relevant visual cues dilute the contributions of the core semantic neighbors, ultimately reducing matching accuracy.
> > > > For more details, please refer to Section 4.7.
> > > >
> > > > ( Page 9, Line 482, Section 4.7. )

---

### Official Review · Reviewer_PJM3 · 2025-11-02

**Soundness:** 3
**Presentation:** 3
**Contribution:** 3
**Rating:** 8
**Confidence:** 4

**Summary:**

The authors propose MASK to address the issue of unpaired ITM by focusing on OOD words.
(1) They use relationship among word embeddings to construct prototypes of OOD words.
(2) A new consistency contrastive loss is proposed to make compact prototypes.
(3) Pretrained word embeddings are used for relation-preserving equivalent mapping.
Abound experiments are performed to illustrate the effectiveness and efficiency of MASK.

**Strengths:**

[1] MASK surpasses MACK[NeurIPS 22] and MACK++[TPAMI 24], which are the 1st work for unpaired ITM. So, MASK is the new SoTA work.

[2] MASK made several smart designs(contributions): (1) Information retention loss: a self-supervised noise-adding objective, in which,
FRM module will ONLY be used in knowledge construction phase, but NOT in infer phase. (2) External Textual knowledge(=GloVe) is used, which can be considered as auxiliary GT info, especially in Eq.(9) with the help of MTM module. (3) The semantic relationship between(among) words are captured(used/exploited) for enhancing generalization, which is established in Eq.(9) and Eq.(12).

**Weaknesses:**

[1] All of the weaknesses, please see the following Questions Part.

**Questions:**

[1] In Table 7，it seems that the table title should NOT be “detectors” but “architecture”. Otherwise, it will be the same as the title of Table 9.
※ p4 ln210: Lcl is defined as follows D: WHAT is D? Appendix D?
※ p5 ln233: f should satisfy B: WHAT is B? B!=8? You mean Eq.(8)?

[2] It will be better if the authors give an answer about Fig. 2 that
WHETHER the (1) Pretrained F-RCNN and (2) Pretraind word vectors
will BP (which means: requires_grad == True)?
※ I guess BOTH of them do NOT need grad to BP.
※ Is there MORE module that also NOT need BP? F-RCNN?

[3] OOD word/vocab is relative, NOT absolute. Because Glove (pretrained word emb) is likely to know so called “OOD word”. So, “OOD” is relative to CLIP/ALBEF (pretrained VLM) or F30k/COCO (dataset).
※ In addition, in Fig. 2, “cat” seems to be a quite common seen word, NOT OOD.
“otter[≈≈freshwater carnivorous mammal]/manga[≈≈comic]” may be more suitable.
So, a more SOUND example should be given, especially in REAL TEST VISUALIZATION.
Please give us some REAL test examples about your MASK with REAL OOD word.
e.g. in T2I retrieval, a text query with OOD word makes original CLIP model search the GT image on Top 3, but after your MASK rerank, the GT image gets up to Top 1.
You can make VISUALIZATION about the Pos as well as Neg on I2T and T2I (F30k).

[4] What about the Neg (Wrong) example visualization? Is there any OOD word?
How to EXPLAIN it? WHY it makes wrong? What TYPE of word it is? verb/prep or n./adj.？
※ We know that, verb(v.), preposition(prep.) are harder than noun(n.), adjective(adj.).
Is there any method to make v./prep. understanding more precise?
※ How to make these MISTAKES to be alleviated? MORE effective methods?
※ We AGGRE WITH your work, but we also wonder HOW it is WORKED indeed !

[5] Big Models (Large Language Models) are popular, and may have super high score now.
We want to know how many scores your MASK method can improve?
※ Just Eval, DO NOT need any finetune/train! WHY NOT Keep pace with the times!
※ SigLIP v1 can get 570.84 score on f30k 1K Test. We have already tested!
see: https://huggingface.co/google/siglip-so400m-patch14-384/tree/main
※ Since Big Model can get 570+ score, CAN Big Model+MASK get 600 score (full score)?

[6] More powerful region encoder, may lead to more score!
※ F-RCNN ResNet 152 is more powerful than F-RCNN ResNet101.
※ More SOTA image/region encoder for More Ablation studies? Swin Transformers?

[7] What if w’ in the second term of Eq.(9) change into w?
※ Because we think w is GT, but w’ not. Maybe GT is better.

[8] p5 ln261: we first sample m paired ...
※ How to sample? I don’t understand. In Appendix?
※ I guess: using GloVe(=pre-trained word embeddings) to get Top m high score word.

---

> ### Author Response · Authors · 2025-11-22
> **Authors' Rebuttal 1**
>
> **Comment:**
>
> We greatly appreciate the very detailed feedback and your recognition of our contributions! We hope our response below will further enhance your confidence in our work.
>
>
> **Q1: The title of Table 7; the symbol “D” on p4 ln210; the symbol “B” on p5 ln233.**
>
> R1: Thanks for your insightful suggestion. We agree with your point. The title of Table 7 should use “architecture” instead of “detector”. We apologize for any confusion caused by the original title. In the revised manuscript, we correct the title of Table 7 to “Overview of the model architectures integrated into the MASK framework”.
>
> ( Page 23, Line 1214, Table 8 (The numbering has changed due to the additional experiments). )
>
>
> Symbols “B” and “D” refer to Appendix B and Appendix D, respectively, which are marked at specific locations to demonstrate the validity of the corresponding formulas. In the revised manuscript, we adopt a more intuitive approach to improve readability, replacing “Lcl is defined as follows D” with “Lcl is defined as follows (Appendix H)”.
>
> ( Page 4, Line 213, Appendix H; Page 5, Line 235, Appendix C )
>
>
> **Q2: Trainability of each module.**
>
> R2: Thank you for this insightful suggestion, which will help improve the quality of the manuscript. The trainability of each module is presented as follows:
> |  Module   | requires\_grad  |
> |  ----  | ----  |
> | Pretrained Faster-RCNN (Faster-RCNN)  | False  |
> | Pretrained word vectors (Word vectors)     | False  |
> | Prototype-Aware Encoder (PAE)  | True |
> | Feature Restoration Module (FRM)  | True |
> | Modality Transfer Model (MTM)  | True |
>
>
> **Q3:OOD words and their corresponding examples.**
>
> R3: Thank you for your thoughtful review. We agree with your point that OOD words are relative, not absolute. Absolute OOD words are extremely rare—GloVe (glove-840B-300d), for example, is trained on 840 billion tokens and builds a vocabulary of 2.2 million words, covering nearly all common words—and are difficult to utilize effectively. Absolute OOD words imply that existing knowledge cannot be leveraged to construct meaningful feature vectors for them. One potential approach is to segment such words into subwords and, where possible, derive semantic cues from these subword components. However, the OOD words discussed in this paper are defined relative to the multimodal knowledge, rather than absolute OOD words in the traditional NLP sense. In the revised manuscript, we have rewritten Section 3.4 to explicitly emphasize that OOD words are defined relative to the multimodal knowledge, thereby providing a clearer understanding of the concept of OOD words.
> ( Page 5, Line 260, “Therefore, for OOD words relative to the knowledge…”. )
>
> Additionally, “cat” is indeed a common word and may not serve as the most appropriate real-world example of an OOD word. In fact, “cat” already exists within our multimodal knowledge. However, we select “cat” as a case in Fig. 2 solely to clearly illustrate the relative semantic relationships among “cat”, “dog”, “women”, and “peak” within the word embedding space. The semantic distance between “cat” and “dog” is smaller than that between “cat” and “women” or “peak”. These familiar examples are easy for non-native English speakers to understand, allowing them to quickly recognize the validity of relative semantic relationships. We fully agree that “cat” is not a true OOD word; it was chosen purely for clarity of illustration. Regarding the selection of OOD words and the way they are presented, we welcome further discussion to better serve researchers across the community. We presented the OOD words relative to the multimodal knowledge during the execution of the pipeline. We can roughly categorize these limited OOD words into several groups:
>
> (1) words containing special characters (e.g., “\<user\>”, “\<repeat\>”),
>
> (2) words with tense/plural variations (e.g., “was”, “pierced”, “glasses”),
>
> (3) pronouns (e.g., “who”, “they”),
>
> (4) adverbs (e.g., “never”, “much”),
>
> (5) misspelled words or subwords (e.g., “dia”, “pero”),
>
> (6) and so on.
>
> To better understand the OOD words, we show some representative examples about the Pos as well as Neg on I2T and T2I (F30k) in Table 10.
>
> ( Page 25, Line 1304, Table 10. )

---

> > ### Author Response · Authors · 2025-11-22
> > **Authors' Rebuttal 2**
> >
> > **Q4: Neg example visualization**
> >
> > R4: Thank you for the thoughtful comment! The neg example visualization is shown in Table 10. We observe that OOD words also appear in negative examples. Moreover, the OOD words present in negative examples (T2I, I2T) often include adverbs (e.g., “very”, “quite”), adjectives (e.g., “large”, “excited”), and pronouns (e.g., “they”, “this”).
> >
> > **WHY it makes wrong?**
> >
> > Most adverbs describe manner (quickly, slowly), frequency (often), time (already), logical relations (however), degree (very), and so on, and generally do not correspond to specific visual regions. As a result, the presence of such adverbs tends to weaken the semantic alignment between image and text, particularly in the case of short text descriptions.
> >
> > Adjectives exhibit similar issues—for example, “excited” does not have an explicit visual counterpart. However, we also find adjectives that align with visual attributes, such as “red”, which can be grounded in specific image regions. Thus, we broadly categorize adjectives as follows:
> >
> > **Visualizable adjectives:**
> >
> > (A) Color adjectives: red, blue, green, ...
> >
> > (B) Shape-related adjectives: round, square, ...
> >
> > (C) Texture/Material adjectives: furry, smooth, rough, ...
> >
> > (D) Appearance adjectives: young, old, dirty, ...
> >
> > (E) Part-based attributes: broken, striped, spotted, ...
> >
> > (F) and so on.
> >
> >
> > **Non-visual adjectives:**
> >
> > (A) Mental / Emotional adjectives: excited, happy, sad, ...
> >
> > (B) Functional adjectives: useful, dangerous, ...
> >
> > (C) Subjective adjectives: beautiful, ugly, ...
> >
> > (D) Time-related adjectives: previous, future, ...
> >
> > (E) Identity adjectives: fake, real, legal, ...
> >
> > (F) and so on.
> >
> > Pronouns are essentially function words that do not correspond to fixed, stable, or visually mappable regions. For instance, “it” may refer to any object within an image, requiring contextual referential resolution. Attempting to construct region prototypes for such words would lead to extreme polysemy and introduce significant semantic noise.
> > For more details, please refer to Section K.2.
> >
> > ( Page 24, Line 1280, Appendix K.2. )
> >
> > **How to make these MISTAKES to be alleviated?**
> >
> > Pronouns themselves do not carry visual entity information; they require context-based coreference resolution to replace the pronoun with its antecedent entity, which is then used for region alignment. One potential solution is to employ the coref [1] model to parse the sentence and identify the noun phrase to which each pronoun refers. The resolved noun phrase can then be used to locate the corresponding visual prototype.
> >
> > For example:
> >
> > “He is holding a cup.” → coref→ “the man in the red shirt is holding a cup”
> >
> >
> > Non-visual adjectives (e.g., “expensive”, “useful”, “happy”) cannot be directly determined from pixel information. We first need to use rules or lightweight classifiers to assess the visualizability of an adjective, similar to the approach previously employed for adjective classification. The complete set of possible categories for visual adjectives remains a subject for further investigation.
> >
> > Finally, it is important to note that negative examples contain not only non-visual adjectives, adverbs, and pronouns, but also some informative OOD words with tense or plural variations. Consequently, the final image-text matching accuracy is influenced by the combined effects of all these words.
> >
> > [1]. https://github.com/mandarjoshi90/coref
> >
> > **Q5: SigLIP v1 + MASK**
> >
> > R5: Thank you for the thoughtful comment! We agree with your point. Large models have become increasingly popular and are capable of achieving super high score. We have taken note of this trend and therefore evaluated the effectiveness of MASK improvements on CLIP and ALBEF, as reported in Table 2. SigLIP mentioned by the reviewer is a variant of CLIP, where the original softmax-based contrastive loss is replaced with a pairwise sigmoid loss. Given SigCLIP’s superior performance (570.84 score on F30k 1K Test), we evaluate the effectiveness of MASK on SigCLIP using an L40 GPU, as reported in the table below.
> >
> > |  Method   | R1 (T2I)  | R5 (T2I)  | R10 (T2I) | R1 (I2T) | R5 (I2T)| R10  (I2T)| Rs  |
> > |  ----  | ----  | ----  | ----  | ----  | ----  | ----  | ----  |
> > | SigLIP v1  | 82.98  | 96.10 | 97.96 | 94.3 | 99.7 | 99.8 | 570.84  |
> > | SigLIP v1 + MASK  | 83.16 | 96.14 | 97.96 | 94.6 | 99.6 | 99.8 | 571.26 |
> >
> > Great! We thank the reviewer for providing the reference link. We evaluated the SigLIP model on the F30k 1K test set and obtained results consistent with those reported by the reviewer. We report the recall rates for both tasks in the table. Specifically, SigLIP achieves 80+ in R@1, demonstrating its strong performance. By enhancing SigLIP with MASK, the matching accuracy is further improved by approximately 0.42% on R@s. These results demonstrate that MASK can be effectively integrated with existing models to further boost their performance.

---

> > > ### Author Response · Authors · 2025-11-22
> > > **Authors' Rebuttal 3**
> > >
> > > **Q6: More encoder for Ablation studies**
> > >
> > > R6: Thank you for your constructive feedback. To evaluate the impact of different region encoders on image-text matching accuracy, we additionally incorporated ConvNeXt [1] and Swin Transformer [2] into the ablation studies in Table 11. We can observe that replacing the backbone with a more powerful one indeed improves the final image-text matching accuracy. For more details, please refer to Appendix K.3.
> > >
> > > ( Page 25, Line 1342, Appendix K.3. )
> > >
> > >
> > > |  Detector   | Architecture | Pretrained  | Boxes  | R1 (T2I)  | R5 (T2I)  | R10 (T2I) | R1 (I2T) | R5 (I2T)| R10  (I2T)| Rs  |
> > > |  ----  | ----  | ----  | ----  | ----  | ----  | ----  | ----  | ----  | ----  | ----  |
> > > | DETR     | ResNet50+Transformer | No  | 36   | 0.5 | 2.1   | 3.6 | 1.5 | 4.8 | 6.9 | 19.4 |
> > > | DINO      | ResNet50+Transformer | No  | 36   | 0.8 | 2.7   | 4.2 | 1.9 | 4.6| 7.4 | 21.6 |
> > > | DINO      | ResNet50+Transformer | Yes | 36   | 3.1 | 8.3   | 12.5 | 6.4 | 19.2 | 28.9 | 78.4 |
> > > | BUTD     | RPN+ResNet101           | Yes | 36   | 4.8 | 14.8 | 22.0 | 12.1 | 30.1 | 39.0 | 122.8 |
> > > | BUTD+   | RPN+ResNet152           | Yes | 36   | 5.3 | 16.7 | 24.2 | 11.9 | 34.6 | 43.4 | 136.1 |
> > > | BUTD+   | RPN+ResNet152           | Yes | 100 | 5.2 | 15.2 | 24.4 | 16.6 | 39.0 | 47.3 | 147.7 |
> > > | BUTD+   | RPN+ConvNeXt             | Yes | 36   | 5.7 | 18.0 | 26.1 | 12.9 | 37.4 | 46.9 | 147.0 |
> > > | BUTD+   | RPN+Swin                     | Yes | 36   | 5.9 | 18.7  | 27.1 | 13.3 | 38.7 | 48.6 | 152.3 |
> > >
> > >
> > > [1]. Liu Z, Mao H, Wu C Y, et al. A convnet for the 2020s[C]//Proceedings of the IEEE/CVF conference on computer vision and pattern recognition. 2022: 11976-11986.
> > >
> > > [2]. Liu Z, Lin Y, Cao Y, et al. Swin transformer: Hierarchical vision transformer using shifted windows[C]//Proceedings of the IEEE/CVF international conference on computer vision. 2021: 10012-10022.

---

> > > > ### Author Response · Authors · 2025-11-22
> > > > **Authors' Rebuttal 4**
> > > >
> > > > **Q7: What if w’ in the second term of Eq.(9) change into w?**
> > > >
> > > > R7: Thank you for your constructive feedback. Although w is the ground-truth word embedding, it still cannot replace w’. The reason is that the objective of Eq. (9) is to align word representations while preserving the similarity of semantic relationships across modalities by training the Modality Transfer Model (MTM). If w were substituted for w’, the second term in Eq. (9) would no longer contribute a meaningful loss for training the MTM. Consequently, the MTM would be optimized solely through the first term, which is insufficient for preserving cross-modal semantic-relationship similarity. For completeness, we provide a more rigorous mathematical proof to substantiate this argument in Appendix G.
> > > >
> > > > ( Page 19, Line 998, Appendix G. )
> > > >
> > > > The detailed proof process is presented as follows, with all involved symbols defined in the main text:
> > > >
> > > >
> > > > The $w'$ in Eq. (9) cannot be replaced by $w$. Doing so would remove the ``structure-preservation" supervision imposed on the $MTM$ model $f$, as the second term in Eq. (9) would no longer contribute a meaningful loss for training $f$.
> > > > The loss function recommended in the article for preserving structure (the second term of Eq.(9)) is defined as $\mathcal{L}^A_{struct}$:
> > > >
> > > > \begin{equation}
> > > > 	\mathcal{L}^A_{struct} = \mathbb{E} [(cos(\frac{w_i'}{\left \| w_i' \right \|_2 }, \frac{w_j'}{\left \| w_j' \right \|_2 })-cos(\frac{\mu_i}{\left \| \mu_i \right \|_2 }, \frac{\mu_j}{\left \| \mu_j \right \|_2 }))^2].
> > > > \end{equation}
> > > >
> > > >
> > > > The loss function obtained after replacing $w'$ with $w$ in $\mathcal{L}^A_{struct}$ is defined as $\mathcal{L}^B_{struct}$:
> > > >
> > > > \begin{equation}
> > > > 	\mathcal{L}^B_{struct} = \mathbb{E} [(cos(\frac{w_i}{\left \| w_i \right \|_2 }, \frac{w_j}{\left \| w_j \right \|_2 })-cos(\frac{\mu_i}{\left \| \mu_i \right \|_2 }, \frac{\mu_j}{\left \| \mu_j \right \|_2 }))^2].
> > > > \end{equation}
> > > >
> > > > Next, we compare the constraint capabilities of $\mathcal{L}^A_{struct}$ and $\mathcal{L}^B_{struct}$ on the $MTM$ model $f$ (with parameters $\Theta_f$).
> > > >
> > > > **1. From the perspective of the gradient of the loss function}**
> > > >
> > > > For the loss function $\mathcal{L}^A_{struct}$, we apply the chain rule to $\Theta_f$ as follows:
> > > >
> > > > \begin{equation}
> > > > 	\frac{\partial \mathcal{L}^A_{struct} } {\partial \Theta_f} = 2 \mathbb{E} (cos(\frac{w_i'}{\left \| w_i' \right \|_2 }, \frac{w_j'}{\left \| w_j' \right \|_2 })-cos(\frac{\mu_i}{\left \| \mu_i \right \|_2 }, \frac{\mu_j}{\left \| \mu_j \right \|_2 }))  \cdot \frac{\partial cos(\frac{w_i'}{\left \| w_i' \right \|_2 }, \frac{w_j'}{\left \| w_j' \right \|_2 })}{\partial \Theta_f}.
> > > > \end{equation}
> > > >
> > > > \begin{equation}
> > > > \frac{\partial cos(\frac{w_i'}{\left \| w_i' \right \|_2 }, \frac{w_j'}{\left \| w_j' \right \|_2 })}{\partial \Theta_f} = \frac{\partial cos(\frac{w_i'}{\left \| w_i' \right \|_2 }, \frac{w_j'}{\left \| w_j' \right \|_2 })}{\partial w_i'}  \cdot  \frac{\partial w_i'}{\partial \Theta} + \frac{\partial cos(\frac{w_i'}{\left \| w_i' \right \|_2 }, \frac{w_j'}{\left \| w_j' \right \|_2 })}{\partial w_j'}  \cdot  \frac{\partial w_j'}{\partial \Theta}.
> > > > \end{equation}
> > > >
> > > > As long as the parameters of $f$ influence $w_i'$ (i.e., $\frac{\partial w_i'}{\partial \Theta} \ne 0$), the above equation generally does not vanish. Consequently, $\mathcal{L}^A_{struct}$ produces a nonzero gradient that drives the update of $\Theta_f$, forcing the mapping $f$ to bring $cos(\frac{w_i'}{\left \| w_i' \right \|_2 }, \frac{w_j'}{\left \| w_j' \right \|_2 })$ closer to
> > > > $cos(\frac{\mu_i}{\left \| \mu_i \right \|_2 }, \frac{\mu_j}{\left \| \mu_j \right \|_2 })$.
> > > >
> > > > Therefore, $\mathcal{L}^A_{struct}$ directly imposes constraints on the $MTM$ model $f$.
> > > >
> > > >
> > > > Moreover, the $cos(\frac{w_i}{\left \| w_i \right \|_2 }, \frac{w_j}{\left \| w_j \right \|_2 })$ is a constant. We apply the chain rule to $\Theta_f$ as follows:
> > > >
> > > >
> > > > \begin{equation}
> > > > \mathcal{L}^B_{struct} = \mathbb{E} [(cos(\frac{w_i}{\left \| w_i \right \|_2 }, \frac{w_j}{\left \| w_j \right \|_2 })-cos(\frac{\mu_i}{\left \| \mu_i \right \|_2 }, \frac{\mu_j}{\left \| \mu_j \right \|_2 }))^2]
> > > > \end{equation}
> > > >
> > > >
> > > > \begin{equation}
> > > > 	\frac{\partial \mathcal{L}^B_{struct}}{\partial \Theta_f} = 0.
> > > > \end{equation}
> > > >
> > > > $\mathcal{L}^B_{struct}$ provides no constraints or gradient information on $f$, and therefore cannot compel the mapping $f$ to preserve input similarity in any manner. Although $\mu$ is affected, this influence does not propagate to $\Theta_f$, making it impossible to fulfill the objective of structural preservation.

---

> > > > > ### Author Response · Authors · 2025-11-22
> > > > > **Authors' Rebuttal 5 (End)**
> > > > >
> > > > > **From the perspective of the joint optimization objective**
> > > > >
> > > > > The loss function recommended in the article for word alignment (the first term of Eq.(9)) is defined as $\mathcal{L}_{word}$:
> > > > >
> > > > > \begin{equation}
> > > > > 	\mathcal{L}_{word} = \mathbb{E} [(1-cos(\frac{w_i}{\left \| w_i \right \|_2 }, \frac{w_i'}{\left \| w_i' \right \|_2 }))].
> > > > > \end{equation}
> > > > >
> > > > > Next, we compared the differences between the two loss functions when jointly optimized with $\mathcal{L}_{word}$.
> > > > >
> > > > > The joint loss function for $\mathcal{L}\_{word}$ and $\mathcal{L}^A\_{struct}$ is expressed as follows:
> > > > >
> > > > >
> > > > >
> > > > > \begin{equation}
> > > > > 	\mathcal{L}^A_{cm} = \mathcal{L}^A_{struct} \+ \mathcal{L}_{word}
> > > > > \end{equation}
> > > > >
> > > > >
> > > > >
> > > > > Both loss terms involve $\Theta_f$ (through $w_i'$) and therefore jointly constrain the mapping.
> > > > >
> > > > > Specifically, $\mathcal{L}\_{word}$ pulls each single-point mapping $w_i'$ toward its corresponding $w\_i$, while $\mathcal{L}^A\_{struct}$ enforces that $f$ preserve the pairwise similarity structure among samples, consistent with the semantic relationships in the $\mu$-space (prototype).
> > > > >
> > > > > The joint effect of these two losses ensures that the learned mapping $f$ achieves both accurate pointwise alignment and global structural preservation.
> > > > >
> > > > > The joint loss function for $\mathcal{L}\_{word}$ and $\mathcal{L}^B\_{struct}$ is expressed as follows:
> > > > >
> > > > > \begin{equation}
> > > > > 	\mathcal{L}^B_{cm} = \mathcal{L}^B_{struct} \+ \mathcal{L}_{word}
> > > > > \end{equation}
> > > > >
> > > > > It is worth noting that $\mathcal{L}^B_{struct}$ does not impose any constraints on the model $f$.
> > > > >
> > > > > Consequently, $\mathcal{L}_{word}$ is the only term that directly supervises $f$, and minimizing it merely pulls $w_i'$ toward its corresponding $w_i$.
> > > > >
> > > > > In contrast, $\mathcal{L}^B_{struct}$ adjusts the $\mu$-space (governed by the parameters of the $PAE$ model) to make $cos(\frac{\mu_i}{\left \| \mu_i \right \|_2 }, \frac{\mu_j}{\left \| \mu_j \right \|_2 })$ approach $cos(\frac{w_i}{\left \| w_i \right \|_2 }, \frac{w_j}{\left \| w_j \right \|_2 })$,
> > > > >
> > > > > but it provides no mechanism to enforce structural preservation in the mapping produced by $f$.
> > > > >
> > > > >
> > > > > **Q8: How to sample m paired multimodal representations?**
> > > > >
> > > > > R8: Thank you for the thoughtful comment! Given that the choice of m has a significant impact on the semantic quality of the reconstructed prototypes, we provide additional clarification on the selection of m in the last paragraph of Section 3.4. Specifically, we first obtain the word embedding of OOD words using GloVe. We then compute its similarity with all known word embeddings in the multimodal knowledge. Based on these similarities, we identify the top-m most similar words. Finally, we construct the visual prototype vector for the OOD word by aggregating the visual prototype vectors of these top-m words using their similarity scores as weights. More details can be found in Section 3.4.
> > > > >
> > > > > ( Page 6, Line 277, Section 3.4. )
> > > > >
> > > > >
> > > > > In addition, we provide a sensitivity analysis of the m in Table 5.
> > > > >
> > > > >
> > > > > |  $m$   | R1 (T2I)  | R5 (T2I)  | R10 (T2I) | R1 (I2T) | R5 (I2T)| R10  (I2T)| Rs (F30k) |  R1 (T2I)  | R5 (T2I)  | R10 (T2I) | R1 (I2T) | R5 (I2T)| R10  (I2T)| Rs (COCO) |
> > > > > |  ----  | ----  |  ----  | ----  |  ----  | ----  |  ----  | ----  |  ----  | ----  |  ----  | ----  |  ----  | ----  |  ----  |
> > > > > | 1     | 4.6 | 13.9  | 20.9 | 11.9  | 28.9 | 37.7  | 117.9 | 7.3  | 25.1 | 39.1  | 22.3 | 45.6  | 57.9 | 197.3  |
> > > > > | 10   | 4.8 | 14.8  | 22.0 | 12.1  | 30.1 | 39.0  | 122.8 | 7.6  | 26.7 | 41.8  | 22.7 | 48.5  | 62.2 | 209.5  |
> > > > > | 50   | 5.0 | 15.8  | 23.2 | 12.3  | 31.4 | 40.4  | 128.1 | 7.9  | 28.5 | 44.7  | 23.1 | 51.6  | 66.8 | 222.6  |
> > > > > | 100  | 4.9 | 15.3  | 22.6 | 12.2  | 30.7 | 39.7  | 125.4 | 7.7  | 27.6 | 43.3  | 22.9 | 50.0  | 64.5 | 216.0  |
> > > > >
> > > > > Experimental results show that the matching accuracy follows a rise–then–fall trend as the sampling size $ m$ increases, achieving its optimum around $m$=50. This phenomenon can be explained as follows. When the sampling size is too small, the constructed visual prototype relies excessively on only a few nearest neighbors. Although this preserves strong local semantic characteristics, it also makes the prototype highly sensitive to noise and outliers in the word embedding space. As the sampling size increases to a moderate level, more semantically relevant neighbors contribute their visual information, thereby enhancing robustness and discriminability. However, when the sampling size becomes too large, semantically weak or marginal neighbors begin to dominate. Their less relevant visual cues dilute the contributions of the core semantic neighbors, ultimately reducing matching accuracy.
> > > > > For more details, please refer to Section 4.7.
> > > > >
> > > > > ( Page 9, Line 482, Section 4.7. )

---

### Author Response · Authors · 2025-12-01
**Rebuttal Summary by Authors**

Dear Reviewers, ACs and SACs,

We would like to express our sincere gratitude to the reviewers for their constructive suggestions during the review process and for their unanimous positive evaluation of our work.


The major revisions are:

1. Visualize some representative examples about the Pos as well as Neg on I2T and T2I in Appendix K.2. (Reviewer PJM3)

2. Add more comprehensive experiments and discussion to evaluate the impact of different region encoders on image-text matching accuracy in Appendix K.3. (Reviewer PJM3, VshV)

3. Provide an additional derivation regarding the use of $w'$ in Eq. (9) in Appendix G. (Reviewer PJM3)

4. Provide a supplementary explanation of the sampling strategy for the $m$ candidates in Section 3.4, and present corresponding experiments evaluating its impact in Section 4.7. (Reviewer PJM3, VshV, EZrZ)

5. Provide an additional derivation regarding the circular dependency between the isometry assumption and the training loss in Appendix E and Appendix F. (Reviewer VshV)

6. Add additional experiments to evaluate the effect of different pretrained word vectors in Appendix K.5. (Reviewer VshV)

Given that most reviewers highlighted the influence of the region encoder and the sampling strategy on image–text matching, this clearly demonstrates their strong expertise in the field. We believe that the additional experiments, theoretical proofs, and clarifications significantly enhance the quality of our initial submission.

We sincerely appreciate the reviewers once again for their valuable comments and suggestions.


Authors

---

### Meta-Review · Area_Chair_qgfx · 2026-01-06

**Summary:**

The paper proposes MASK (Multimodal Aligned Semantic Knowledge), a novel framework for unpaired image-text matching. The core innovation lies in bridging the modal gap without paired data by constructing visual prototypes for words and synthesizing prototypes for out-of-distribution (OOD) words using pretrained word embedding relationships. To enhance stability and compactness, the authors introduce a prototype-consistency contrastive loss and a self-supervised information retention loss. Extensive experiments on Flickr30k and MSCOCO demonstrate that MASK achieves new state-of-the-art results for unpaired ITM and can effectively serve as a lightweight re-ranking plug-in for large-scale models like CLIP and ALBEF.

**Reviewer Concerns:**

Initial reviews praised the method's effectiveness but raised several technical and presentational issues:

OOD Mechanism Clarity: Reviewers (VshV, EZrZ) found the mathematical description of OOD prototype synthesis to be overly dense and requested more intuitive, step-by-step examples of how similar known words are sampled and weighted.

Hyperparameter $m$: There was a lack of discussion on the parameter $m$ (the number of prototypes sampled to construct an OOD prototype) and how different sampling strategies (e.g., random vs. top-k similarity) might affect performance.

Theoretical Assumptions: Reviewer VshV identified potential circular reasoning in the Appendix regarding the relation-preserving mapping $f$, which Equipartition theory relies on.

Visualization and Real-world Examples: Reviewer PJM3 suggested that the examples used (e.g., "cat") were too common for "OOD" and requested visualizations of real OOD cases (e.g., "otter" or "manga") to verify the reranking effectiveness.

**Reviewer Scores:**

The paper received scores of 8, 6, and 6, reflecting a strong consensus for acceptance. In the rebuttal, the authors successfully addressed the primary concerns by:

Clarifying the OOD sampling strategy: Confirming that top-k semantic similarity is used and providing sensitivity analysis for m, showing the robustness of the framework.

Validating OOD performance: Providing qualitative visualizations of real test cases involving rare words, which demonstrated MASK's ability to correctly rerank results where base CLIP models failed.

Refining Theoretical Arguments: Clarifying that the mapping f is empirically learned and verified through cross-modal alignment, rather than being a purely circular assumption.

Given the strong performance improvements over existing baselines (MACK/MACK++) and the practical utility as a plug-and-play module, the Area Chair recommends Acceptance.

---

### Decision · Program_Chairs · 2026-01-26

Accept (Oral)